# Specific heterozygous variants in *MGP* lead to endoplasmic reticulum stress and cause spondyloepiphyseal dysplasia

Ophélie Gourgas[1,11], Gabrielle Lemire[2,3,4,11], Alison J. Eaton[2,5], Sultanah Alshahrani[6,7], Angela L. Duker [8], Jingjing Li [1], Ricki S. Carroll [8], Stuart Mackenzie[8], Sarah M. Nikkel [9], Care4Rare Canada Consortium*, Michael B. Bober[8], Kym M. Boycott[2,3] & Monzur Murshed [1,6,10] ✉

Matrix Gla protein (MGP) is a vitamin K-dependent post-translationally modified protein, highly expressed in vascular and cartilaginous tissues. It is a potent inhibitor of extracellular matrix mineralization. Biallelic loss-of-function variants in the *MGP* gene cause Keutel syndrome, an autosomal recessive disorder characterized by widespread calcification of various cartilaginous tissues and skeletal and vascular anomalies. In this study, we report four individuals from two unrelated families with two heterozygous variants in *MGP*, both altering the cysteine 19 residue to phenylalanine or tyrosine. These individuals present with a spondyloepiphyseal skeletal dysplasia characterized by short stature with a short trunk, diffuse platyspondyly, midface retrusion, progressive epiphyseal anomalies and brachytelephalangism. We investigated the cellular and molecular effects of one of the heterozygous deleterious variants (C19F) using both cell and genetically modified mouse models. Heterozygous 'knock-in' mice expressing C19F MGP recapitulate most of the skeletal anomalies observed in the affected individuals. Our results suggest that the main underlying mechanism leading to the observed skeletal dysplasia is endoplasmic reticulum stress-induced apoptosis of the growth plate chondrocytes. Overall, our findings support that heterozygous variants in *MGP* altering the Cys19 residue cause autosomal dominant spondyloepiphyseal dysplasia, a condition distinct from Keutel syndrome both clinically and molecularly.

Skeletal dysplasias are a group of clinically and genetically heterogeneous disorders associated with abnormal bone and cartilage development. These genetic conditions are clinically characterized by short stature with abnormal skeletal proportions and morphology, which typically result from dysregulation of pathways involved in patterning, bone growth and mineralization[1]. The latest version of the Nosology and Classification of Genetic Skeletal Disorders from the International Skeletal Dysplasia Society includes 461 clinically defined

[1]Department of Medicine, McGill University, Montreal, QC, Canada. [2]Children's Hospital of Eastern Ontario Research Institute, University of Ottawa, Ottawa, ON, Canada. [3]Department of Genetics, Children's Hospital of Eastern Ontario, Ottawa, ON, Canada. [4]Broad Center for Mendelian Genomics, Program in Medical and Population Genetics, Broad Institute of MIT and Harvard, Cambridge, MA, USA. [5]University of Alberta, Edmonton, AB, Canada. [6]Faculty of Dental Medicine and Oral Health Sciences, McGill University, Montreal, QC, Canada. [7]Department of Oral and Maxillofacial Surgery, Faculty of Dentistry, King Abdulaziz University, Jeddah, Saudi Arabia. [8]Nemours Children's Health, Wilmington, DE, USA. [9]University of British Columbia, Vancouver, BC, Canada. [10]Shriners Hospitals for Children - Canada, Montreal, QC, Canada. [11]These authors contributed equally: Ophélie Gourgas, Gabrielle Lemire. *A list of authors and their affiliations appears at the end of the paper. ✉e-mail: monzur.murshed@mcgill.ca

skeletal dysplasias, with the molecular etiologies being established in 425 of these disorders[2]. While the clinical aspects and molecular mechanisms of some forms of skeletal dysplasias have been well delineated, there remain others for which the underlying causes are still unknown, and previously unrecognized conditions within this group continue to emerge.

In the current study, we report four individuals from two unrelated families with previously undescribed spondyloepiphyseal dysplasia and heterozygous variants in the matrix Gla protein gene (*MGP*) affecting residue Cys19. MGP is a small, secreted protein that acts as a potent inhibitor of extracellular matrix (ECM) mineralization in the vascular and cartilaginous tissues. It belongs to a family of γ-carboxylated glutamic acid (Gla) containing proteins and has been classified as a skeletal Gla protein[3–5]. The 5 glutamic acid residues in MGP undergo γ-carboxylation by gamma glutamyl carboxylase (GGCX) in a vitamin K-dependent manner. In addition, three N-terminal conserved serine residues are phosphorylated by yet unknown kinase(s)[4,6]. These post-translational modifications are thought to be critical for MGP's anti-mineralization function[7].

Biallelic deleterious variants resulting in loss of function of MGP cause Keutel syndrome (KS) (OMIM #245150), an autosomal recessive disorder characterized by abnormal cartilage and vascular tissue calcification, midfacial hypoplasia, peripheral pulmonary artery stenosis, brachytelephalangism, mild developmental delay and hearing loss[8–11]. Gene expression analyses using in situ hybridization demonstrated strong *Mgp* expression in the vascular smooth muscle cells and chondrocytes in mice[12]. Consistent with this expression pattern, widespread ectopic mineralization (calcification) was detected in the arterial and cartilaginous tissues of MGP-deficient (*Mgp*^−/−^) mice[5]. These mice faithfully recapitulate many of the clinical features seen in individuals with KS, but with a more severe vascular calcification phenotype[5,13]. Analyses of *Mgp*^−/−^ mice and individuals with KS over the past two decades have confirmed that the skeletal abnormalities are primarily associated with apoptosis of the chondrocytes and ectopic calcification of the cartilaginous tissues in the skeleton[11,13–15].

While KS is an autosomal recessive disorder, we suspected that the skeletal dysplasia described in this report was caused by heterozygous variants in *MGP* and followed an autosomal dominant mode of inheritance. Considering that the Cys19 variants alter an amino acid in the signal peptide, which is normally cleaved during the cotranslational translocation process, we hypothesized that mature MGP would retain the mutated signal peptide and the resultant modified protein would have altered functional properties leading to the skeletal abnormalities. In this study, we provide experimental evidence to support this hypothesis in both cell culture and mouse models.

Here we show that our mouse model expressing the mutant protein reproduces the spondyloepiphyseal dysplasia phenotype observed in the affected individuals. We demonstrate that the processing of the signal peptide in the mutated C19F MGP is impaired leading to its accumulation in the endoplasmic reticulum (ER). This, in turn, causes ER stress and apoptosis of chondrocytes affecting the normal development of the endochondral bones. Our work identifies potential cellular and molecular mechanisms underlying the skeletal dysplasia caused by heterozygous Cys19 variants in *MGP*.

## Results
### Clinical description of affected individuals from Family 1 and 2
Individuals 1, 2, and 3 from Family 1 presented with short stature with a disproportionate short trunk, short hands, mild midface retrusion, and epiphyseal anomalies (Fig. 1a, b, c, Table 1). The 52-year-old mother from this family, Individual 1, presented with progressive epiphyseal degeneration leading to hip pain and functional limitations, which required bilateral hip replacement at 33 years of age. Individual 4 from Family 2 presented with short stature with a disproportionate short trunk, short hands, rhizomelia, exaggerated lumbar lordosis, midface

retrusion and epiphyseal anomalies (Fig. 1a, d, e, and Table 1). There were no other affected individuals in Family 2. In the four affected individuals, X-rays identified diffuse platyspondyly with biconcavity of the middle column of the vertebrae, giving each vertebral body a pisces-like appearance in the lateral view (Fig. 1c, e). Radiographs of the hands were notable for brachytelephalangism of the lesser digits, which was also apparent clinically. The pelvis was proportionate with capacious acetabuli. Ossification of the acetabuli was irregular and patchy in the skeletally immature patients. The lower extremities featured broad, flattened epiphyses of the femur and tibia, most pronounced in the proximal femurs. In this small group of affected individuals, coxa valga and genu valgum were present in bilateral lower extremities. The linear growth of Individual 4 was noted to have plateaued after 10 years of age, and a workup for endocrine causes of decreased growth velocity was normal. X-rays revealed closure of the distal tibial growth plates, which was unexpected given that Individual 4 was premenarchal (Supplementary Fig. 1, A–F). Individual 4 also presented with severe corneal ulceration/destruction, which required corneal transplant and was suspected to have been caused by a viral infection. Please see the Supplementary Data 1 for detailed phenotypic information.

### Exome sequencing identified heterozygous variants in *MGP* in Family 1 and 2
Exome sequencing analysis identified a heterozygous missense variant in *MGP* that was shared among the three affected individuals from family 1 (NM_000900.3:c.56G>T:p.C19F). This variant is absent from gnomAD[16] and thus rare, and in silico prediction tools predict the change of this conserved residue to impact protein structure and function (CADD score of 27). Sanger sequencing confirmed that the variant in *MGP* was present in the heterozygous state in the affected mother and her two affected children (Individuals 1, 2 and 3), and absent from the unaffected father and the unaffected sibling. No other variants in known or novel genes have been retained as plausible candidates by exome analysis. Interrogation of the ClinVar database for interesting variants in *MGP* identified a heterozygous variant affecting MGP's Cys19 residue that had been submitted by a clinical laboratory. This ClinVar submission pertained to the proband from Family 2 (Individual 4), in whom exome sequencing analysis had identified a de novo heterozygous missense variant in *MGP* (NM_000900.3:c.56G>A:p.C19Y). Subsequent communication with the clinical team involved in the care of this patient revealed an overlapping phenotype and genotype between the two families. The c.56G>A variant is absent from gnomAD [16] and thus rare, and in silico prediction tools predict the change of this conserved residue to impact protein structure and function (CADD score of 25). No other variants in known or novel genes have been retained as plausible candidates in the exome analysis of Individual 4. The two single nucleotide substitutions, c.56G>T and c.56G>A, in the *MGP* gene alter a conserved Cys19 residue in the signal peptide and lead to phenylalanine (F) and tyrosine (Y) substitution, respectively.

### CRISPR/Cas9-mediated mutagenesis to introduce the C19F variant in MGP in mice
Of the two variants affecting the Cys19 codon (56G>T and 56G>A), we examined the effects of the 56G>T (C19F) on cell fate and the underlying regulatory pathways. We replaced the codon TGC to TTC in a "knock-in" mouse model using a CRISPR/Cas9-based mutagenesis approach. The guide RNA-Cas9 complex together with the single-stranded oligodeoxynucleotide (ssODN) repair template carrying the replacement phenylalanine anti-codon (complementary sequence for the ssODN is shown in Fig. 2a) were microinjected into the male pronucleus of murine fertilized eggs, which were then transferred to pseudo-pregnant females.

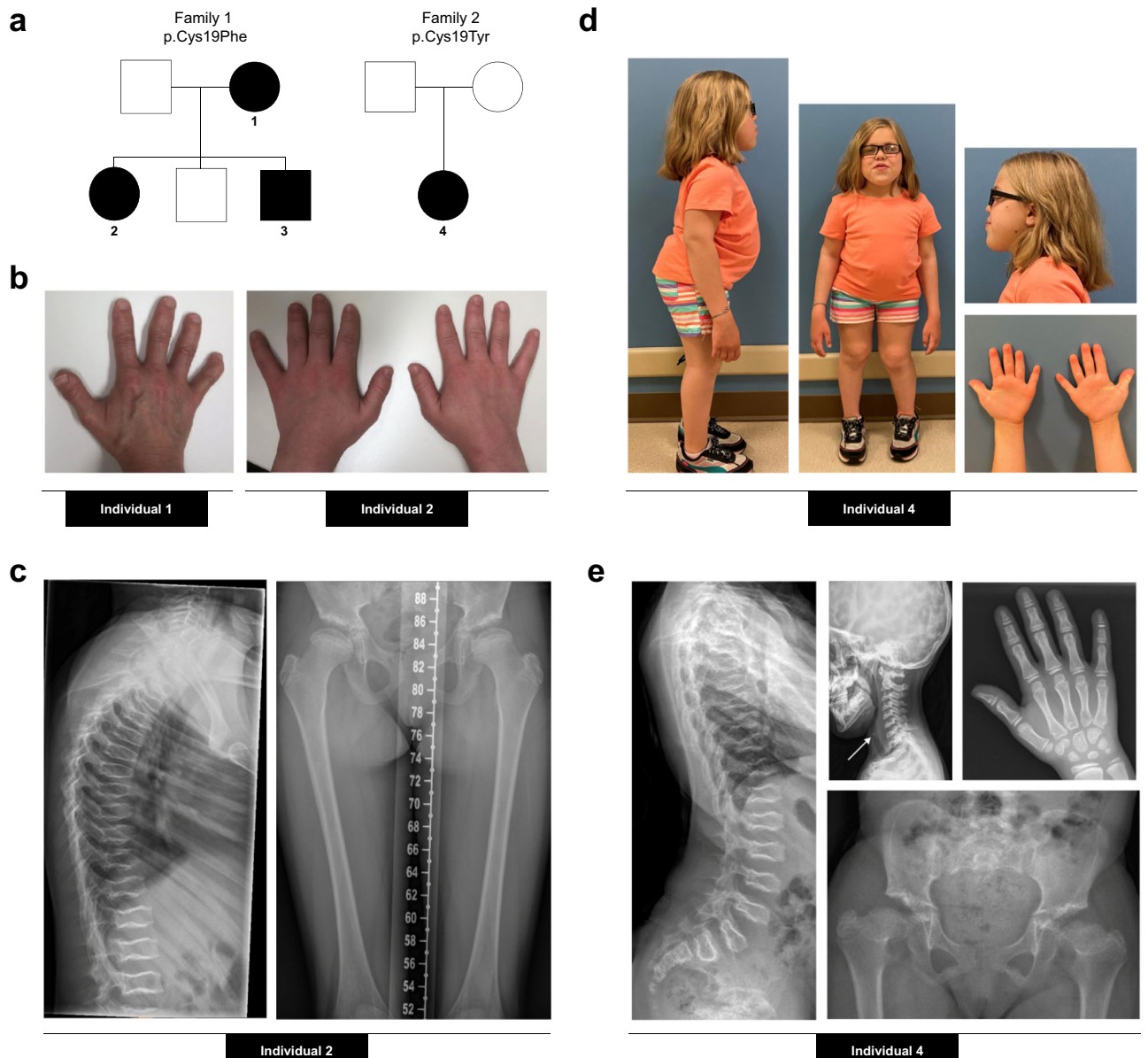

**Fig. 1 | Clinical description of affected individuals from Family 1 and 2.**
**a** Pedigrees of the two affected families. Black symbols represent affected individuals. **b** Photographs of the hands of two individuals from Family 1 showing short hands (including short palms and brachydactyly) and brachytelephalangism. Left: Individual 1 (at 52 years of age); Right: Individual 2 (at 18 years of age). **c** X-rays of Individual 2 from Family 1 at 10 years of age showing diffuse platyspondyly with biconcave vertebral bodies throughout spine, flattened and broad bilateral femoral epiphysis with patchy ossification of the acetabular roof. **d** Photographs of Individual 4 from Family 2 at 10 years of age, showing short stature with a disproportionate short trunk, rhizomelia, exaggerated lumbar lordosis, midface retrusion, short hands with brachydactyly and mild bilateral genu valgum. **e** X-rays of Individual 4 from Family 2 at 7 years of age showing diffuse platyspondyly with biconcave vertebral bodies throughout spine, broad, flattened bilateral proximal femoral epiphyses, coxa vara and brachytelephalangism of the lesser fingers. X-ray of lateral cervical spine shows premature calcification of the cricoid cartilage (white arrow).

The presence of the desired variant in mice generated by microinjections was examined by PCR of the target genomic sequence followed by Sanger sequencing of the PCR amplicons. A total of 4 of the mice generated by microinjection were of short stature and their DNA analyses showed the presence of the mutated locus. A representative image of electrophoretic analysis of the PCR products indicating the presence of alterations in the *Mgp* loci is presented in Fig. 2b. The 400 base pair band was expected to carry the wild type (WT) and/or mutated sequence, while the high molecular weight bands suggest about the insertion events at one of the *Mgp* loci. Sanger sequencing of the 400 base pair band (Fig. 2c) shows that one founder (F0 male) carried the *56G>T* variant (C19F). Other mice with similar skeletal phenotypes showed the same point mutation with additional mutations at the *Mgp* loci. Our repeated attempts to breed the F0 male were unsuccessful; thus, we proceeded for sperm collection to generate F1 heterozygous mice by in vitro fertilization (IVF). Sanger sequencing confirmed the presence of one mutated and one normal allele in F1 heterozygous mice (Fig. 2d) generated by IVF. These mice were used for phenotypic analyses presented in the subsequent Figures. A table is included showing the total number of mice generated which carried the *56G>T* variant and its association with the observed skeletal phenotypes (Fig. 2e).

Next, we performed sequencing of the cDNAs using the *Mgp*-specific primers which showed that our DNA manipulation methods did not introduce any "off-target" mutations in the MGP coding sequence in the mutant mice (Supplementary Fig. 2). Further, we used

**Table 1 | Clinical features and body measurements of the four affected individuals**

| | Family 1 | | | Family 2 |
|---|---|---|---|---|
| Individual | 1 | 2 | 3 | 4 |
| *MGP* variant (NM_000900.3) | c.56G>T/ p.Cys19Phe (C19F) | | | c.56G>A/ p.Cys19Tyr (C19Y) |
| Sex | Female | Female | Male | Female |
| Age at last clinical evaluation (years) | 52 | 18 | 12 | 10 |
| Height (SD) | 148.8 cm (−2.2 SD) | 140.3 cm (−3.4 SD) | 133.2 cm (−2.2 SD) | 106.4 cm (−5SD) |
| Armspan/Height ratio | 1.06 | 1.08 | 1.05 | 1.09 |
| Upper-to-lower segment ratio | 0.93 | 0.84 | 0.9 | 1.14[a] |
| Sitting height (SD) | 70 cm (<2SD) | 68.5 cm (< 2 SD) | 62 cm (< 2 SD) | 58.1 cm (<2SD) |
| Platyspondyly | + | + | + | + |
| Short hands and brachydactyly | + | + | + | + |
| Rhizomelia | − | − | − | + |
| Midface retrusion | + | + | + | + |
| Brachytelephalangism | + | + | + | + |
| Femoral head epiphyseal dysplasia | + | + | + | + |

[a]The upper to lower segment ratio of Individual 4 has to be interpreted in the context of the patient's complex lower extremity deformities and pelvic tilt, which result in a falsely decreased symphysis to floor measurement.

a web-based application (CRISPOR) to examine whether our gRNA might have targeted any other sequences in the mouse genome to cause off-target effects. One potential region (GRCm39) was identified, and the possible target DNA in one of the $Mgp^{+/56G>T}$ mice generated by IVF (F1 generation) was amplified using specific primer pairs flanking the region. No alteration of the target DNA sequence was observed by Sanger sequencing (Supplementary Fig. 3).

All the mutant mice were normal at birth. As shown by micro-CT imaging, $Mgp^{+/56G>T}$ mice at 1 week and 2 weeks of age had a comparable normal length of long bone (tibia) to the WT littermates (Fig. 2f). We measured the body weights of $Mgp^{+/56G>T}$ males and females and compared to that of their sex-matched control littermates. All $Mgp^{+/56G>T}$ mice had a markedly reduced weight gain after 2 weeks of age (Fig. 2g). At 6 weeks of age, $Mgp^{+/56G>T}$ mice showed an overall shortening of the body length with a smaller size compared to the sex-matched control mice (Fig. 2h). All the mutant mice died before 2 months of age.

### C19F variant in MGP results in shorter bones, midface retrusion, and osteopenia in mice

The X-ray images of 6-week-old $Mgp^{+/56G>T}$ mice carrying the C19F variant in MGP revealed that the overall size of their skeleton was smaller with shorter vertebrae and decreased radio opacity in various bones (Fig. 3a). Measurements of femoral and tibial lengths of 6-week-old $Mgp^{+/56G>T}$ mice showed a significant shortening of the limbs compared to those of the control mice (Fig. 3b). X-ray images of the hip joints of $Mgp^{+/56G>T}$ mice showed hypermineralized cartilaginous tissue in the ilium, adjacent to the acetabulum (Fig. 3c). Micro-CT analysis of the $Mgp^{+/56G>T}$ mice's cranium at 6 weeks of age showed craniofacial anomalies including midface retrusion and a severely reduced bone mass in both intramembranous and endochondral bones. Micro-CT images also revealed abnormal calcification of the tracheal cartilage in the mutant mice (Fig. 3d). Interestingly, X-ray imaging did not show any sign of thoracic aorta calcification in $Mgp^{+/56G>T}$ mice (Fig. 3a), which was also confirmed by alizarin red staining (Supplementary Fig. 4).

The low bone mass phenotype in endochondral bones was further confirmed by the micro-CT analyses of the lumbar vertebrae and distal femur of the mutant mice (Fig. 3e–h). There was a significant reduction of the trabecular bone volume over tissue volume (BV/TV), trabecular number (calculated as the inverse of the average distance between the midlines of each adjacent trabecula) and thickness and an increase of trabecular spaces in the lumbar vertebrae of $Mgp^{+/56G>T}$ mice (Fig. 3e, f). We also observed a reduction of trabecular bone mass in the distal femur of the $Mgp^{+/56G>T}$ mice (Fig. 3g). Indeed, there was a significant decrease of BV/TV and trabecular number as well as a significant increase of trabecular spaces in the distal femur of these mice (Fig. 3h). However, we found a very mild difference in the cortical bone mass between the mutant and control mice (Fig. 3h).

### C19F variant in MGP results in poor bone remodeling in mice

We next prepared histological sections of the plastic embedded lumbar vertebrae from 6-week-old mice and stained them by von Kossa (stains mineralized matrices) and van Gieson (stains unmineralized collagen) (Fig. 4a) as described before[14,17]. In agreement with our micro-CT data, we found a significantly lower trabecular BV/TV in $Mgp^{+/56G>T}$ mice. Interestingly, we found wider intervertebral disks with larger annulus fibrosus and nucleus pulposus in the vertebral joints of the mutant mice. Our histomorphometric analysis showed a 67% decreased trabecular bone volume and matching reduction of trabecular number in the mutant mice compared to the sex-matched littermates (Fig. 4b).

To investigate the cause of the low bone mass phenotype in $Mgp^{+/56G>T}$ mice, we examined the dynamic parameters affecting bone remodeling. We measured the lumbar vertebrae mineral apposition rate (MAR) and bone formation rate over bone surface (BFR/BS) using double calcein labeling in 6-week-old $Mgp^{+/56G>T}$ and control mice. We found a slight decrease in both MAR and BFR/BS in the mutant mice although these differences were not statistically significant (Fig. 4c, d). Toluidine blue staining of histological sections showed that the osteoblast number was significantly reduced in $Mgp^{+/56G>T}$ mice over the tissue area (Ob/T.Ar) or bone perimeter (Ob/B.Pm) (Fig. 4e, f). We next counted the number of tartrate resistant acid phosphatase (TRAP)-positive osteoclasts on $Mgp^{+/56G>T}$ and control bone sections. The osteoclast numbers over tissue area (Oc/T.Ar) and over bone perimeter (Oc/B.Pm) in $Mgp^{+/56G>T}$ mice were significantly lower than in the control ones (Fig. 4g, h). Overall, these data suggest that there is a poor bone remodelling in the mice expressing the C19F variant.

### Growth plate abnormalities in $Mgp^{+/56G>T}$ mice

We next examined the growth plate phenotypes in $Mgp^{+/56G>T}$ mice. We performed von Kossa and safranin O (VKSO; stains mineralized matrices and proteoglycans) staining of the undecalcified lumbar vertebral sections of the control and $Mgp^{+/56G>T}$ mice at 6 weeks of age. As expected, this dual staining showed the presence of continuous unmineralized cartilaginous matrices in the vertebral growth plates of WT mice (Fig. 4i). In $Mgp^{+/56G>T}$ mice, however, the growth plates appeared thinner with very little or no unmineralized cartilage. A comparison with the vertebral growth plates of $Mgp^{-/-}$ mice and recently reported $Mgp^{S3mut/S3mut}$ mice lacking the N-terminal serine residues that undergo post-translational phosphorylation showed that the severe growth plate phenotype is unique to $Mgp^{+/56G>T}$ mice. Both $Mgp^{-/-}$ mice and $Mgp^{S3mut/S3mut}$ mice showed ectopic calcification disrupting the continuity of the unmineralized growth plates, however, the phenotype is markedly milder than $Mgp^{+/56G>T}$ mice at 6 weeks of age. We next decalcified the vertebral sections of 6-week-old control and $Mgp^{+/56G>T}$ mice in EDTA overnight and stained by VKSO, which showed reduced amount of cartilage matrix and severe disruption of the cellular organization of the growth plates in $Mgp^{+/56G>T}$ mice (Fig. 4j). The tibial growth plates in $Mgp^{+/56G>T}$ mice showed similar abnormalities, although less severe; in some areas, growth plates were

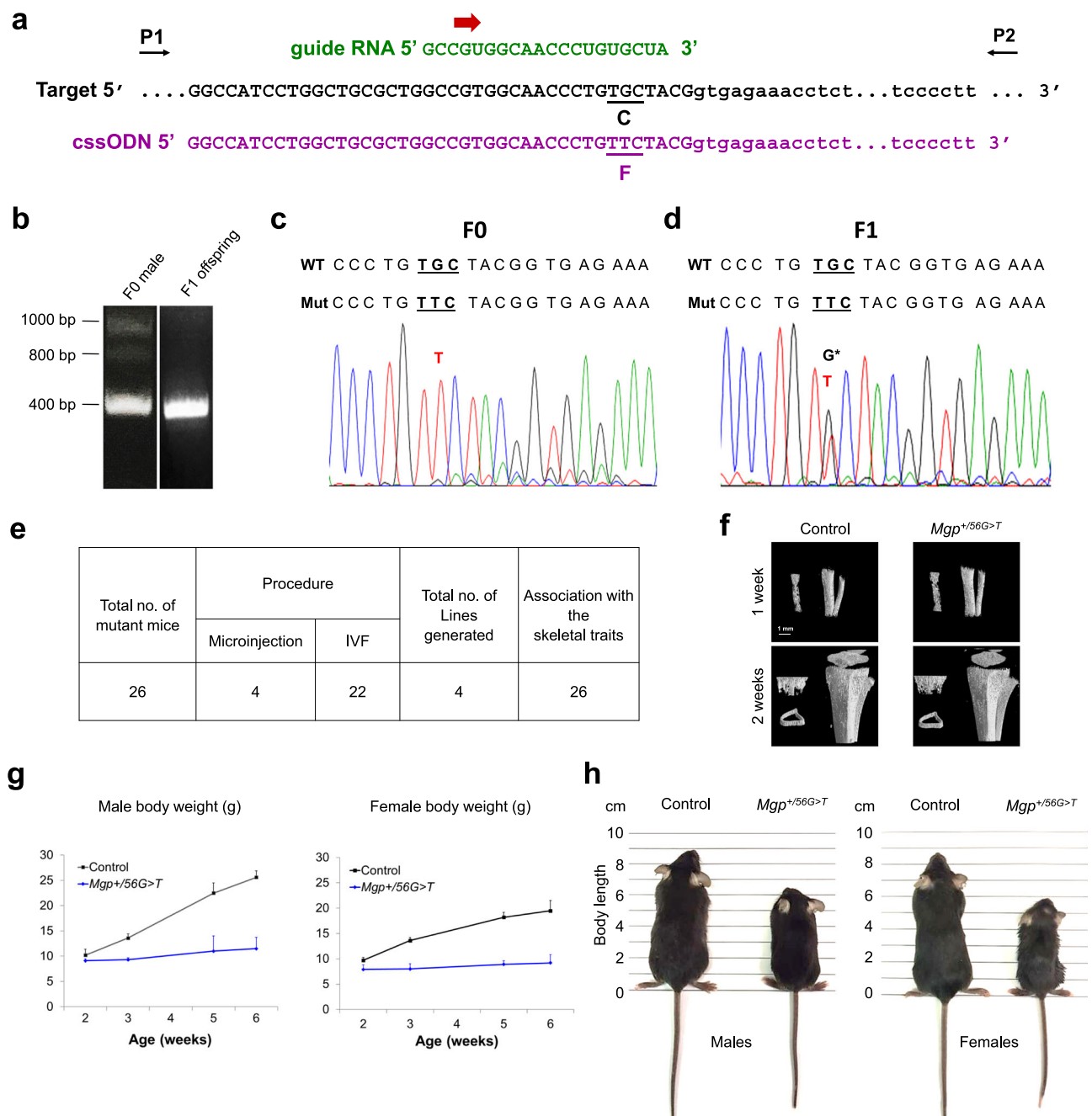

**Fig. 2 | CRISPR/Cas9-mediated mutagenesis to introduce the C19F variant in MGP in mice. a** *Mgp* target sequence, sequence of the validated guide RNA, and the sequence complementary to the single-stranded oligodeoxynucleotide (cssODN) used to introduce the desired C19F variant. The original cysteine codon in the target sequence and the replacement phenylalanine codon are underlined and marked as C and F, respectively. The ssODN carries homology arms flanking each side of the mutated nucleotide. The flanking arrows show the primer pair (P1 and P2) used for the genotyping PCR and DNA sequencing. PCR using these primers would yield amplicons of ~400 bps band after agarose gel electrophoresis. **b** A representative agarose gel image of electrophoresed PCR products showing amplicons generated from a F0 male and its F1 offspring carrying the desired C19F variant. The agarose gel electrophoresis of PCR amplicons generated from the F0 male DNA shows multiple PCR bands−a lower band with the expected size (~400 bps) as well as high molecular weight bands. We predicted that the high molecular weight bands were due to the presence of insertion mutations in one of the *Mgp* alleles. PCR genotyping of a F1 male using the same primer pair shows the lower band only most likely because different *Mgp* alleles in the F0 male have been segregated in the offspring. **c.** Sanger sequencing of the lower band shows the presence of the mutated allele (c.56G>T), but not the WT allele in the F0 male further supporting the inference that the other *Mgp* locus is altered by insertion mutations in this mouse. **d** Similar PCR analysis followed by DNA sequencing show the presence of both the WT and the mutated sequences in a heterozygote F1 mouse. Note the peak corresponding to the guanine nucleotide (G*) overlapping the thymine (T) peak in the chromatogram for the F1 mouse, but not for the F0 mouse. **e** Table showing the total number of *Mgp*$^{+/56G>T}$ mice generated from multiple microinjection and IVF experiments. **f** Micro-CT images of the tibia of 1- and 2-week-old control and *Mgp*$^{+/56G>T}$ mice. Note, the skeletal development is comparable between these two genotypes. **g** Body weight measurements of control and *Mgp*$^{+/56G>T}$ male and female mice from 2–6 weeks of age. *Mgp*$^{+/56G>T}$ mice exhibit poor weight gain after 2 weeks of age. Error bars represent standard deviations; *n* = 3–8 mice for each group at each time point. Data presented as mean ± SD. Source data are provided as a Source Data file. **h** Photographs of control and *Mgp*$^{+/56G>T}$ male and female mice at 6 weeks of age.

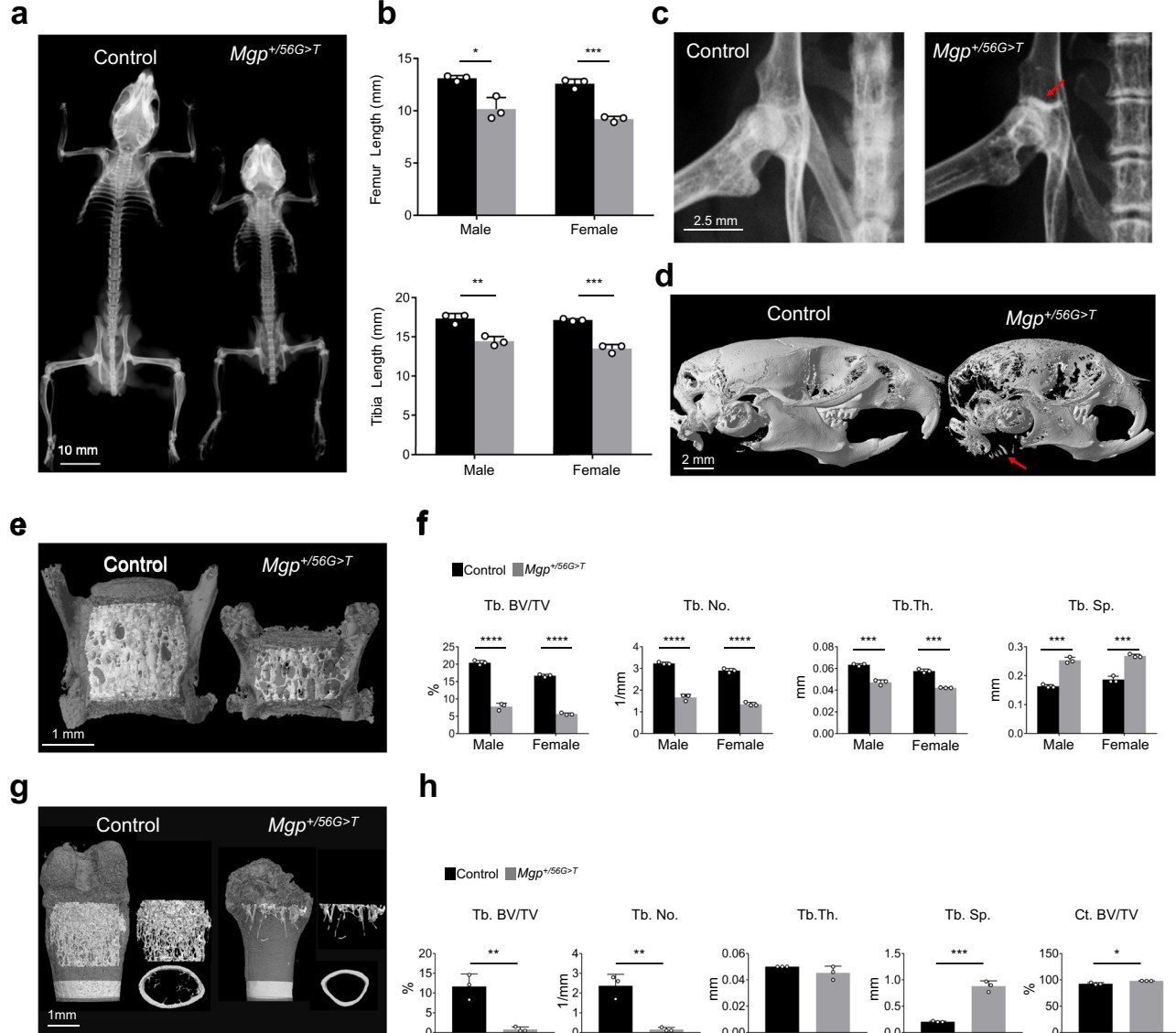

**Fig. 3 | The C19F variant in MGP results in shorter bones, midface retrusion and osteopenia. a** X-ray images of a control and $Mgp^{+/56G>T}$ male mice at 6 weeks of age. Heterozygous mice carrying the C19F variant are shorter, show facial anomalies, low bone mass, and shortened vertebrae. **b** Measurements (based on the X-ray images) of the femur and tibia of 6-week-old male and female mice showing significantly shorter long bones in $Mgp^{+/56G>T}$ mice compared to the control mice ($n = 3$ mice/group). **c** X-ray images of the hip joint of a control and $Mgp^{+/56G>T}$ male mice at 6 weeks of age. The red arrow shows the hypermineralized cartilaginous tissue in the ilium, adjacent to the acetabulum, in $Mgp^{+/56G>T}$ mice. **d** Micro-CT images of heads from both control and $Mgp^{+/56G>T}$ male mice at 6 weeks of age showing craniofacial anomalies and midface retrusion, and abnormal calcification of the tracheal cartilage (red arrow) in mutant mice. **e** Micro-CT images of the lumbar vertebrae (L3) from both control and $Mgp^{+/56G>T}$ male mice at 6 weeks of age. The

vertebrae from the $Mgp^{+/56G>T}$ mice are shorter than the control mice. **f** The comparative analysis of the trabecular bones (L3 and L4) showed a significant reduction of trabecular bone volume over tissue volume (Tb. BV/TV), trabecular number (Tb.No.; the average number of trabeculae per mm), trabecular thickness (Tb. Th.) and an increase in the trabecular spacing (Tb. Sp.) ($n = 3$ mice/group). **g** Micro-CT images of 6-week-old control and $Mgp^{+/56G>T}$ distal femur. **h** Quantitative analysis of micro-CT parameters shows a significant reduction of Tb. BV/TV and Tb. No. and an increase in Tb. Sp. However, there is no significant difference in Tb. Th. between control and $Mgp^{+/56G>T}$ distal femur ($n = 3$ mice/group). There is a mild increase of Ct. BV/TV (cortical bone volume over tissue volume) in the mutant mice. Error bars represent standard deviations. Data presented as mean ± SD. *$P < 0.05$; **$P < 0.01$; ***$P < 0.001$; ****$P < 0.0001$ vs. control by two tailed unpaired $t$ test. Actual $P$ values and source data are provided as a Source Data file.

thinner, with fewer hypertrophic chondrocytes or prematurely closed due to abnormally deposited minerals (Supplementary Fig. 5).

To investigate the cause of the abnormal growth plate phenotypes in $Mgp^{+/56G>T}$ mice, we examined the presence of three different growth plate ECM proteins—type II collagen (general chondrocyte marker), aggrecan (marker for prehypertrophic chondrocytes) and type X collagen (marker for hypertrophic chondrocytes) in 3-week-old mice. We found that while the growth plate chondrocytes in mutant mice did not reduce the expression of type II collagen (Fig. 4k) and aggrecan (Fig. 4l), the distribution patterns of these two proteins were altered,

and there was almost a complete loss of the hypertrophic zone with very few chondrocytes expressing type X collagen (Fig. 4m).

### C19F variant in MGP leads to its intracellular accumulation and poor secretion

To investigate how the C19F variant is causing the skeletal phenotypes observed in vivo, we first generated transiently transfected ATDC5 chondrogenic cells expressing either the WT or mutated MGP. Since the currently available anti-MGP antibodies do not provide satisfactory immunodetection, we used plasmid vectors expressing WT or C19F

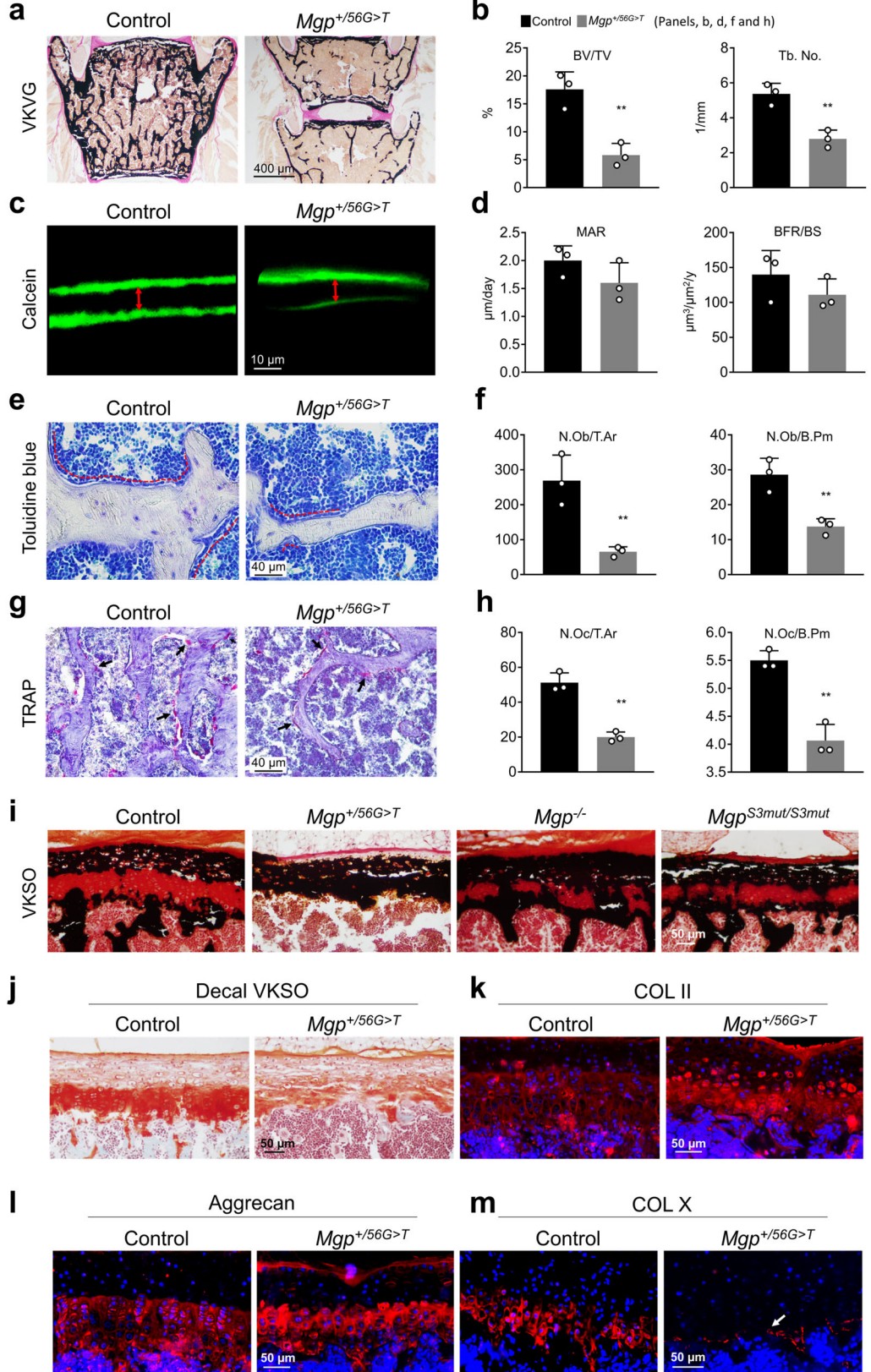

MGP, which were tagged with an epitope tag (FLAG) at the C-terminal end. The p*Mgp-FLAG-IRES-GFP* and p*C19FMgp-FLAG-IRES-GFP* plasmids also produce the jellyfish green fluorescent protein (GFP) as a non-fused reporter protein (Fig. 5a). Upon transfection, comparable expression of GFP in both ATDC5 cultures transfected with the WT or

the C19F MGP expressing plasmids suggested their comparable transfection efficiencies (Fig. 5b). Upon the initial failures to detect the WT or mutated proteins by Western blotting due to poor transfection efficiency and weak expression of the transfected constructs, we performed immunoprecipitation using anti-FLAG magnetic beads to

**Fig. 4 | The C19F variant in MGP results in poor bone remodeling and growth plate abnormalities in mice. a, b** Histomorphometric analysis of lumbar vertebrae sections stained with von Kossa and van Gieson (VKVG) reveals a low bone mass phenotype in 6-week-old $Mgp^{+/S6G>T}$ mice when compared with the control mice (n = 3 mice/group). BV/TV= trabecular bone volume over tissue volume, Tb.No.= trabecular number (the average number of trabeculae per mm). **c, d** The quantification of the calcein double labeling shows no significant reduction of the mineral apposition rate (MAR) and bone formation rate per unit of bone surface (BFR/BS) in the $Mgp^{+/S6G>T}$ mice when compared to the control mice (n = 3 mice /group). The red arrows show the distance between calcein double labels. **e, f** Toluidine blue-stained vertebral sections show significantly reduced osteoblast (dotted red lines) counts in the mutant mice compared to the control mice (n = 3 mice/group). N.Ob/T.Ar = Osteoblast number over tissue area, N.Ob/B.Pm = Osteoblast number over bone perimeter. **g, h** The tartrate resistant acid phosphatase (TRAP)-stained sections reveal significantly reduced osteoclast (arrows) numbers in $Mgp^{+/S6G>T}$ mice (n = 3 mice/group). N.Oc/T.Ar = Osteoclast number over tissue area, N.Oc/B.Pm = Osteoclast number over bone perimeter. Error bars represent standard deviations; Data presented as mean ± SD. **P < 0.01 vs. control by two tailed unpaired t test. Actual P values and source data are provided as a Source Data file. **i.** Von Kossa and safranin O (VKSO) staining of the undecalcified lumbar vertebral sections from the control and $Mgp^{+/S6G>T}$ mice, as well as $Mgp^{-/-}$ and $Mgp^{S3mut/S3mut}$ mice at 6 weeks of age. The growth plates of $Mgp^{+/S6G>T}$ mice are more severely hypermineralized and prematurely closed compared to that of the control, $Mgp^{-/-}$ or $Mgp^{S3mut/S3mut}$ mice. **j** VKSO staining of the decalcified lumbar vertebra sections from 6-week-old control and $Mgp^{+/S6G>T}$ mice. The typical proteoglycan-rich safranin O-stained cartilaginous ECM seen in the control mice is markedly altered in the mutant mice. **k, l, m** Anti-type II collagen, anti-Aggrecan and anti-type X collagen staining of the lumbar vertebra sections from 3-week-old control and $Mgp^{+/S6G>T}$ mice. Note the much thinner hypertrophic zone and markedly reduced type X collagen staining (arrow) in the growth plates of $Mgp^{+/S6G>T}$ mice.

precipitate/pull down FLAG-tagged MGP proteins from the filter-concentrated culture media and cell lysates of transfected ATDC5 cells. Our Western blotting data showed a 17 kDa band in addition to the 15 kDa band in the extracts from ATDC5 cells expressing the mutant protein (Fig. 5c). Also, the amount of the mutated intracellular protein appeared to be more than that of the WT protein. Interestingly, when the same kind of blotting experiments were performed with the immunoprecipitated proteins from the culture media, very little mutant protein was found to be secreted (Fig. 5d). This observation was supported by a separate set of transfection experiments with two new plasmid vectors expressing the WT (pMgp-GFP) or C19F MGP (pC19FMgp-GFP) fused in frame to GFP (Fig. 5e). Fluorescence imaging showed that C19F MGP protein fused to GFP was retained more in the transfected ATDC5 cells (Fig. 5f).

Considering the poor expression of exogenous proteins in the transfected chondrogenic ATDC5 cells, we decided to use human embryonic kidney (HEK-293) cells for further analyses. We used two plasmid vectors in which the WT or C19F mutated MGP coding sequences were fused in frame to a FLAG tag coding sequence at the 3' end (Fig. 5g). Co-transfection of HEK-293 cells by each of these constructs together with a GFP expression vector showed comparable transfection efficiencies (Fig. 5h). Our gene expression analysis by qRT-PCR also showed comparable expression of both native (WT) and mutated (C19F) Mgp in the transfection experiments (Fig. 5i).

Western blotting analyses also identified a higher molecular weight band (-17 kDa) in the C19F MGP expressing cell extracts in addition to the band representing the WT protein (-15 kDa) (Fig. 5j). Densitometric analysis consistently showed higher amounts of C19F MGP protein in the cell lysate compared to that of the WT MGP.

To check whether the mutated protein was normally secreted outside the cells, we next analyzed the culture media collected 48 h post-transfection. As described above, anti-FLAG magnetic beads were added to the filter-concentrated conditioned media to precipitate the tagged proteins. Western blotting of the bound protein showed a marked reduction of mutated MGP in the culture medium suggesting that the C19F variant affects the secretion of the mutated protein (Fig. 5k). These results were further confirmed by immunofluorescence using an anti-FLAG antibody showing that the mutant protein is mostly localized within the cells (Fig. 5l).

## C19F variant in MGP results in impaired processing of the signal peptide

Our mass spectrometry analyses were successful in detecting approximately 67% of MGP peptides in WT Mgp and C19F Mgp expression vector-transfected cell extracts and in the culture media. Overall, we found that the MGP peptides were more abundant in the extracts prepared from HEK-293 cells expressing the mutant form of MGP, while their presence was very low in the culture media. Opposite results were obtained for the extracts or culture media from the cells expressing the WT MGP (Fig. 5m, n).

After searching for all known post-translational modifications (PTMs) using the Mascot error tolerant search, both WT and mutant variants of MGP were found to be comparably modified. Interestingly, using a "no enzyme" or semi-trypsin search, we found that the C19F MGP samples contained several MGP peptides carrying the mutated phenylalanine (F19) residue as well as a few other upstream amino acids present in the signal peptide. In contrast, the WT samples did not show the presence of any MGP peptides carrying the original cysteine (C19) or any upstream residues from the signal peptide (Fig. 5o).

## Transfected cells expressing C19F MGP show increased ER stress and death

Our Western blotting, immunolabelling and mass spectrometric analyses indicated that C19F MGP is accumulated intracellularly, most likely within the endoplasmic reticulum (ER). We next examined whether this intracellular accumulation of the mutant protein resulted in ER stress and abnormal cell death. HEK-293 cells were transfected with the plasmid vectors expressing either the WT or the C19F MGP (both FLAG-tagged). Immunofluorescence analyses showed that two different ER stress markers, calnexin and C/EBP Homologous Protein (CHOP), were both upregulated in cells expressing the FLAG tagged C19F MGP (Fig. 6a, b). Further high-resolution confocal imaging confirmed that the FLAG tagged C19F MGP protein was colocalized with ER protein calnexin (Fig. 6c). In a separate transfection experiment, labelling of dead cells using ethidium homodimer, showed a significant increase in the number of dead cells in C19F MGP expressing cells compared to the WT MGP expressing cells (Fig. 6d). We next studied the nature of cell death in cultures expressing C19F MGP. Using TUNEL assay, we showed that cells with intracellular accumulation of the mutant protein underwent DNA fragmentation, a hallmark of programmed cell death (Fig. 6e).

## C19F variant in MGP results in ER stress and increased apoptosis of chondrocytes in vivo

We next examined whether growth plate chondrocytes in $Mgp^{+/S6G>T}$ mice showed the signs of ER stress recapitulating our in vitro findings. We performed immunofluorescence analyses of the vertebral sections from 3-week-old control and $Mgp^{+/S6G>T}$ mice to detect multiple ER stress markers. Our microscopy data showed that, while calnexin expression in WT mice was at the basal level and evenly distributed within the cytosol of prehypertrophic/hypertrophic cells, in $Mgp^{+/S6G>T}$ mice, more intense and localized accumulation of calnexin was detected in the prehypertrophic/hypertrophic chondrocytes (Fig. 7a).

While increased accumulation of calnexin is one of the first signs of ER stress leading to apoptosis, the subsequent events may include the activation of one or more pathways involving three ER membrane proteins ATF6, PERK and/or IRE1α as depicted in Fig. 7b. We next

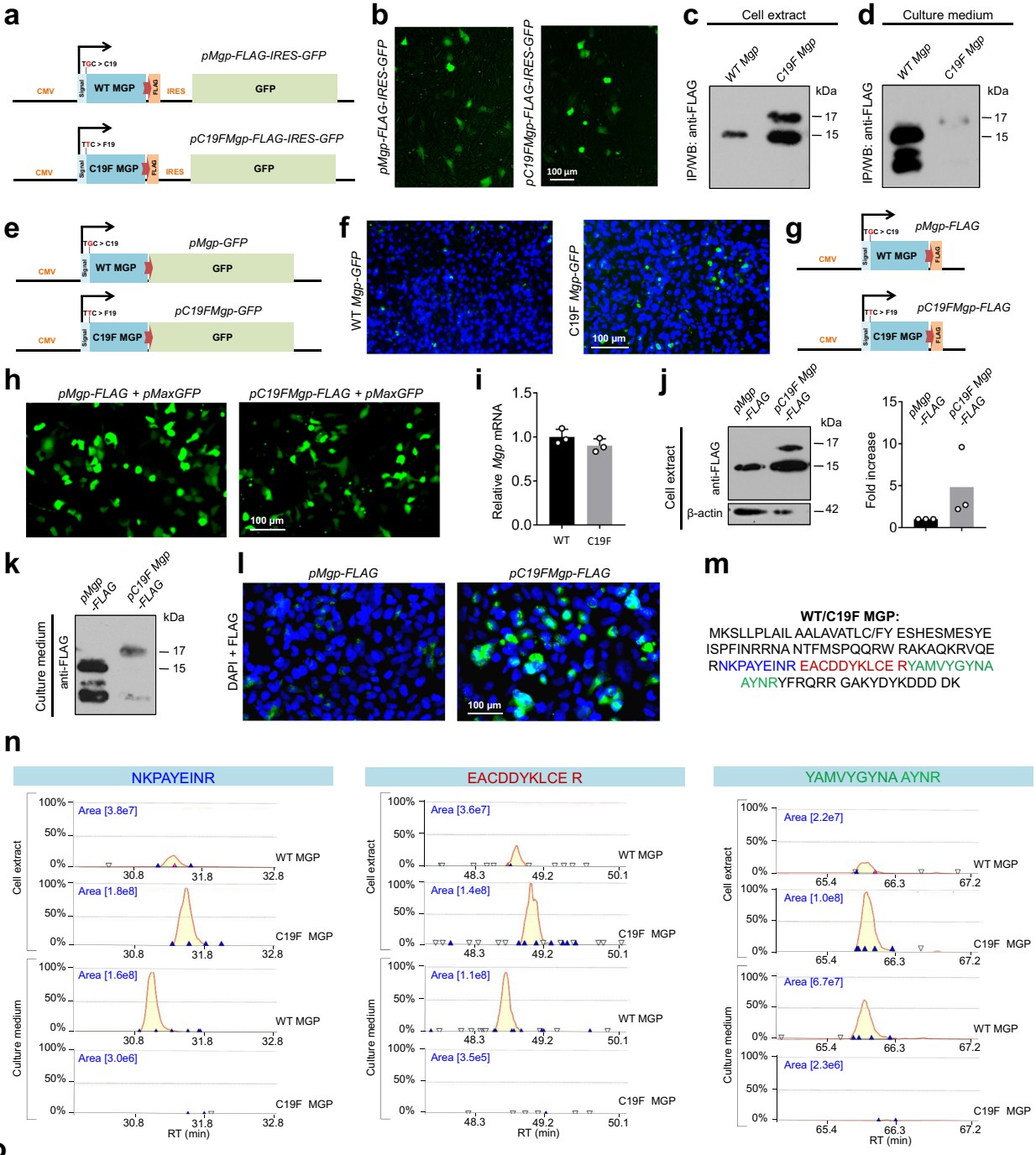

examined the levels and distribution of ATF6 (N), p-eIF2α and spliced form XBP1 (sXBP1), three downstream mediators of ATF6, PERK and IRE1α, respectively in the growth plates of control and $Mgp^{+/S6G>T}$ mice. Immunofluorescence imaging using confocal microscopy showed that ATF6 (N) and p-eIF2α were both increased in the prehypertrophic/hypertrophic cells of the growth plates of $Mgp^{+/S6G>T}$ mice in comparison to that of the control mice (Fig. 7c, d). However, sXBP1 levels were comparable in the growth plates of these two genotypes (Fig. 7e). In agreement with these findings, we found an increased expression of CHOP, a common downstream response protein activated by ER stress pathways, in the growth plate hypertrophic chondrocytes in $Mgp^{+/S6G>T}$ mice (Fig. 7f).

**Fig. 5 | The C19F variant in MGP results in impaired processing of the signal peptide and intrecallular accumulation of the mutated protein.** Impaired processing, intracellular accumulation and poor secretion of C19F MGP in ATDC5 (**a**–**f**) and HEK-293 (**g**–**o**) cells. **a** Scheme showing the *pMgp-FLAG-IRES-GFP* or *pC19FMgp-FLAG-IRES-GFP* plasmids expressing the FLAG-tagged WT or C19F MGP. The constructs also express internal ribosome entry site (IRES)-driven GFP. **b** Fluorescence images showing GFP expression indicate comparable transfection efficiencies by both the plasmids in ATDC5 cells. Cell extracts or culture media from the transfected ATDC5 cells were enriched by immunoprecipitation (IP) using anti-FLAG magnetic beads and subjected to immunoblotting using an anti-FLAG antibody. C19F MGP expressing cells show-17 kDa and -15kDa bands in the extracts (**c**), but not in the culture medium (**d**). **e** Plasmid vectors *pMgp-GFP* and *pC19FMgp-GFP* expressing the WT or C19F MGP fused in frame to GFP. **f** Fluorescence images showing higher retention of C19F MGP-GFP in the transfected ATDC5 cells. **g** Scheme showing the FLAG-tagged WT (*pMgp-FLAG*) or mutated (*pC19FMgp-FLAG*) MGP expressing plasmids. Co-transfection of HEK-293 cells with GFP-expressing *pMaxGFP* and *pMgp-FLAG* or *pC19FMgp-FLAG* plasmids shows comparable transfection efficiency. **i** qRT-PCR showing comparable levels of WT and mutant *Mgp*

RNA in the transfected HEK-293 cells (n = 3 independent biological samples/group). Graph shows mean±SD. Anti-FLAG immunoblots of the cell extracts (**j**) and culture media (**k**) from HEK-293 cells transfected with the *pMgp-FLAG* or *pC19FMgp-FLAG* constructs. Culture media were enriched for the tagged protein by IP as above. C19F MGP expressing cells show the -17 kDa and -15 kDa bands and higher amount of tagged proteins in the extract, but not in the medium. Molecular weights were shown in kilodalton (kDa). Densitometric analysis shows a consistent increase of C19F MGP protein in the cell lysate compared to that of the WT MGP. Graph shows mean of three independent experiments. **l.** Immunofluorescence imaging using the anti-FLAG antibody shows accumulated C19F MGP inside the transfected HEK-293 cells. The nuclei were stained by DAPI (blue). **m** WT/C19F MGP sequence. Peptides examined (n) are shown in color. **n** Representative mass spectrometry analyses of cell extracts or culture media from the transfected HEK-293 cells show more abundant tagged proteins in the extracts expressing C19F MGP, while their amount is very low in the culture medium. Analyses of cells expressing WT MGP result in an opposite pattern. **o** A "no enzyme" or semi-trypsin search shows that the C19F MGP samples contain several MGP peptides carrying the mutated F19 residue, often with some upstream amino acids. Source data are provided as a Source Data file.

Next, to examine whether increased ER stress leads to chondrocyte apoptosis in the growth plates, we performed TUNEL assay. Our data showed that there was a drastic increase in the number of apoptotic chondrocytes mostly in the prehypertrophic/hypertrophic zones of the vertebral growth plates in *Mgp*[+/56G>T] mice at 3 weeks of age (Fig. 8a, b). When the growth plates of 6-week-old mice were analyzed by TUNEL assay, we still observed the patches of hypertrophic cells undergoing apoptosis in *Mgp*[+/56G>T] mice, but not in the control mice (Fig. 8c). Interestingly, no such abnormally high cell deaths were observed in *Mgp*[-/-] or *Mgp*[S3mut/S3mut] mice.

Our final set of experiments were focused on chondrogenic ATDC5 cells to examine whether these cells showed increased cell death upon expression of C19F MGP and whether such deaths could be prevented by treating them with 4-Phenylbutyric acid (4-PBA), a known inhibitor of ER stress. We transfected ATDC5 cells, with plasmid vectors expressing MGP-GFP or C19FMGP-GFP fusion proteins. Upon transfection, 48 h later, cells were stained with ethidium homodimer and both GFP and ethidium homodimer labelled cells were counted. The numbers of double labelled cells over total GFP-positive cells were significantly higher in cultures expressing the mutant protein (Fig. 8d, e). In a separate transfection experiment, when the C19F MGP expressing ATDC5 cells were treated with 5 mM 4-PBA, there was a significant reduction of cell deaths (Fig. 8f). In a separate transfection experiment, we showed that the MGP-GFP fusion protein was retained at a higher amount intracellularly when produced together with C19F MGP, but not with WT MGP suggesting that the mutant protein may interfere with the secretion of the WT protein (Supplementary Fig. 6).

Overall, the data presented above confirmed that the C19F variant led to retention of the signal peptide causing impaired protein secretion, ER stress and premature death in chondrocytes (Fig. 9).

## Discussion

Exome sequencing combined with a one-sided matchmaking strategy resulted in the identification of four individuals from two unrelated families with a heterozygous variant in *MGP* affecting the same highly conserved cysteine (Cys 19) residue. These individuals presented with a strikingly overlapping spondyloepiphyseal dysplasia phenotype, which was distinct from that of individuals with KS caused by biallelic loss-of-function variants in *MGP*[1]. Platyspondyly and progressive epiphyseal degeneration were significant in the four affected individuals in this report which were not described in patients with KS. Also, pulmonary artery stenosis, arterial calcification, hearing loss, and developmental delay are common clinical features in KS and were not present in the individuals from this cohort[8,18–20]. Nonetheless, there were a few notable phenotypic similarities between the current cohort of affected individuals and KS patients. Radiographs of the hands of

affected individuals demonstrated brachytelephalangism of the lesser digits, which is seen in KS. Midface retrusion is also a clinical feature observed in individuals with KS[11,21] and was seen in our cohort, especially in Individual 4 carrying the 56G>A variant. In addition, Individual 4 did appear to have premature calcification of the cricoid cartilage, albeit not as significant as the vast tracheal ring calcifications seen in KS. Short stature has also been reported in individuals with KS[10,18]. Despite these similarities, the defining features of the affected individuals with heterozygous 56G>T or 56G>A variants in *MGP*, the vertebral and epiphyseal anomalies, make the condition clinically distinct from KS. We suggest naming this condition Spondyloepiphyseal dysplasia (SED), *MGP* type. Reporting of additional affected individuals in future will improve the clinical delineation of SED, *MGP* type.

The identification of novel *MGP* variants affecting the same cysteine residue in individuals with a previously unreported skeletal disorder and their autosomal dominant mode of inheritance demanded a thorough investigation into the underlying cell and molecular mechanisms. The substituted cysteine is the last amino acid in MGP's signal sequence, thus not expected to be present in the mature protein. We therefore reasoned that the processing of the mutated signal peptide would be impaired, resulting in a larger protein carrying the unprocessed signal peptide. This inference implied that the mutant protein's localization and functions might be altered. Our in vitro and in vivo experiments provided strong evidence that this is indeed the case.

Considering that *Mgp*[-/-] mice carrying homozygous *Mgp* null variants faithfully recapitulate the features of human KS[5], we decided to investigate the effects of the C19F MGP variant in a mouse model. The recent advances in in vivo genome editing methodologies prompted us to opt for a CRISPR/Cas9-based approach to generate a "knock-in" mouse model. Interestingly, sequencing of the *Mgp* locus-specific PCR amplicons of expected size generated from the F0 male mouse showed only the thymine substitution (56G>T), but not the original guanine nucleotide at position 56. This can be explained by the additional insertion mutations in one of the loci resulting in high molecular weight bands upon electrophoresis, which was later confirmed by our observation that upon IVF using the sperms of the F0 male, differentially mutated *Mgp* alleles were segregated in the resultant offspring.

Despite the ease of genome editing, CRISPR/Cas9-based approach has been shown to introduce undesirable "off-target" mutations[22]. Our DNA sequence analyses showed no "off-target" mutation other than 56G>T in one of the *Mgp* locus. Moreover, since the visual examination of all the *Mgp*[+/56G>T] mice generated from different microinjection and IVF events expressing C19F MGP showed the same physical traits, we concluded that they could not be associated with any particular off-target effects. Nevertheless, we used an online application to examine

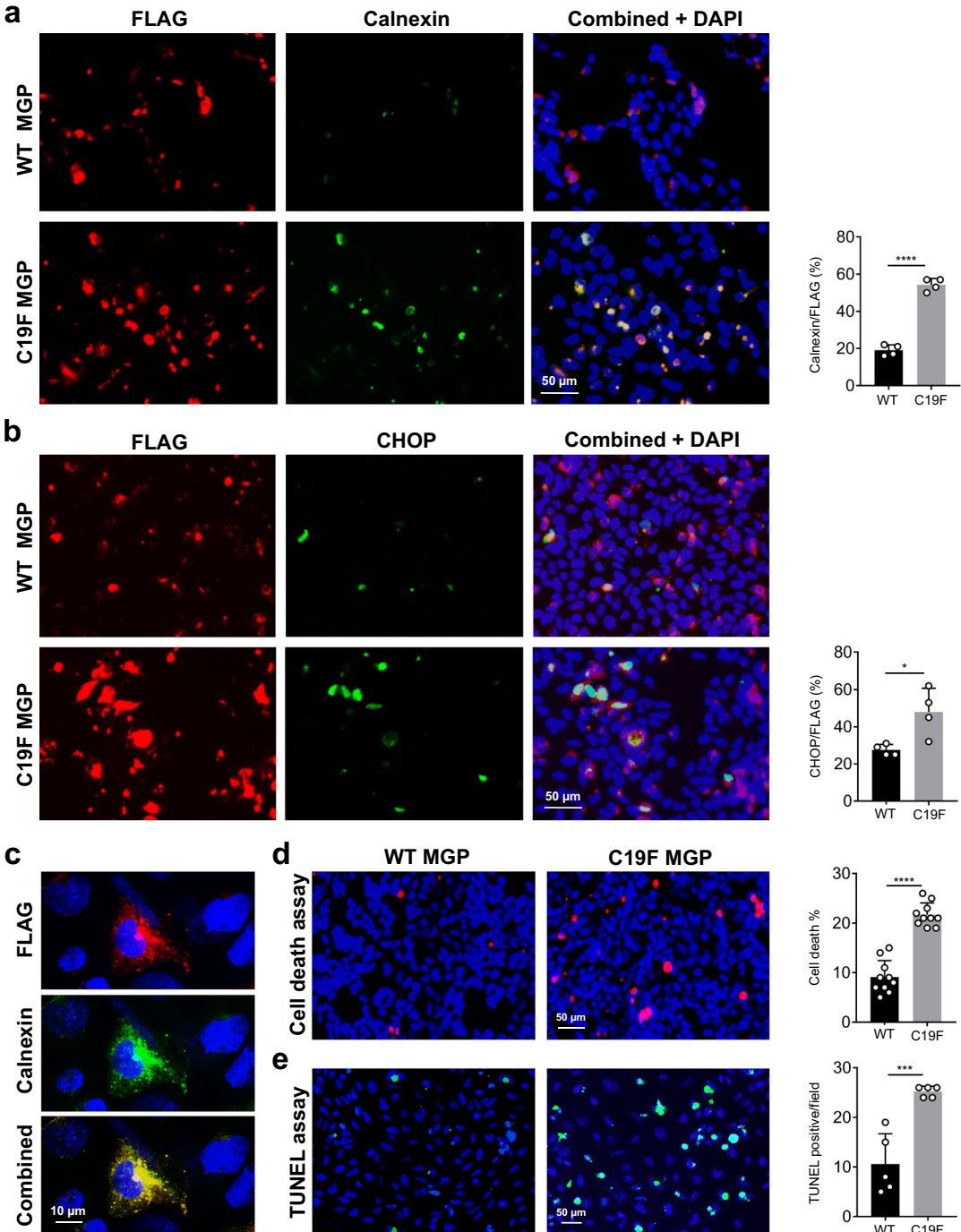

**Fig. 6 | The C19F variant in MGP causes ER stress and cell death in HEK-293 cells. a** Immunofluorescence imaging using anti-FLAG and anti-calnexin antibodies shows the upregulation of calnexin in HEK-293 cells expressing C19F MGP. DAPI stains the nuclei (blue) ($n = 4$ individual image fields/group). **b** Immunofluorescence imaging using an anti-CHOP antibody shows the upregulation of CHOP in cells expressing C19F MGP. DAPI stains the nuclei ($n = 4$ individual image fields/group). **c** Confocal immunofluorescence images of transfected HEK-293 cells stained with anti-FLAG and anti-calnexin antibodies show that FLAG-tagged C19F MGP protein is co-localized in the ER with the stress marker calnexin. **d** Cell death assay. Ethidium homodimer-stained (red) transfected HEK-293 cells show a significant increase in the number of dead cells expressing C19F MGP when compared to those expressing WT MGP ($n = 10$ individual image fields/group). **e** TUNEL assay shows increased apoptosis in C19F MGP expressing cells ($n = 5$ individual image fields/group). Data presented as mean ± SD. *$P < 0.05$; **$P < 0.01$; ***$P < 0.001$; ****$P < 0.0001$ vs. WT by two tailed unpaired $t$ test. Actual $P$ values and source data are provided as a Source Data file.

whether there is any chromosomal sequence that might be targeted by the gRNA used in our experiments. However, sequencing of a potential target area in $Mgp^{+/S6G>T}$ genomic DNA did not identify any alteration of the original sequence.

Since the individuals carrying the C19F variant had platyspondyly, we examined the growth plates, active areas for endochondral bone

development where strong $Mgp$ expression has been reported previously[12]. Although rodents maintain unmineralized growth plates beyond the first year of life, we found the tibial and vertebral growth plates in the $Mgp^{+/S6G>T}$ mice narrower and prematurely mineralized already by 6 weeks of age. Premature mineralization of the tibial physes with decreased growth velocity was also observed in one

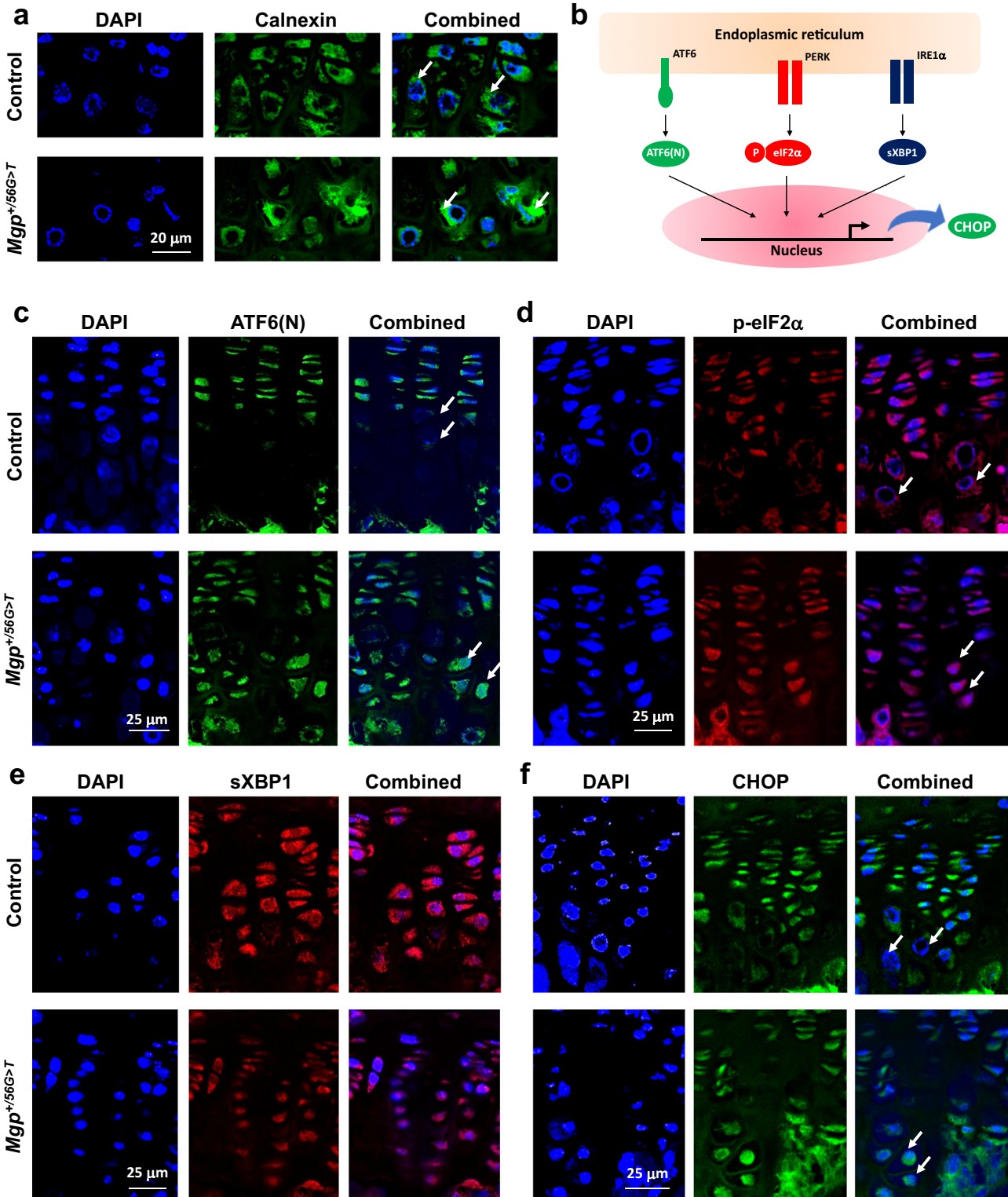

**Fig. 7 | The C19F variant in MGP results in ER stress in vivo.**
**a** Immunofluorescence imaging using an anti-calnexin antibody shows the upregulation of calnexin in the prehypertrophic and hypertrophic chondrocytes in the growth plates of 3-week-old *Mgp*^+/56G>T^ mice. DAPI stains the nuclei (blue).
**b** Schematic representation of three major ER stress pathways involving ATF6, PERK and IRE1α and their downstream effectors N-terminal fragment of ATF6

(ATF6-N), phosphorylated eIF2α (peIF2α) and the spliced form of XBP1 (sXBP1), respectively. Confocal immunofluorescence images of vertebral sections from 3-week-old control and *Mgp*^+/56G>T^ mice using anti-ATF6-N (**c**), anti-p-eIF2α (**d**), anti-sXBP1 (detects the spliced form) (**e**), and anti-CHOP antibodies (**f**) show increased accumulation of these markers, except for the spliced XBP1 in the latter genotype. Arrows on the immunofluorescence images indicate the protein localization.

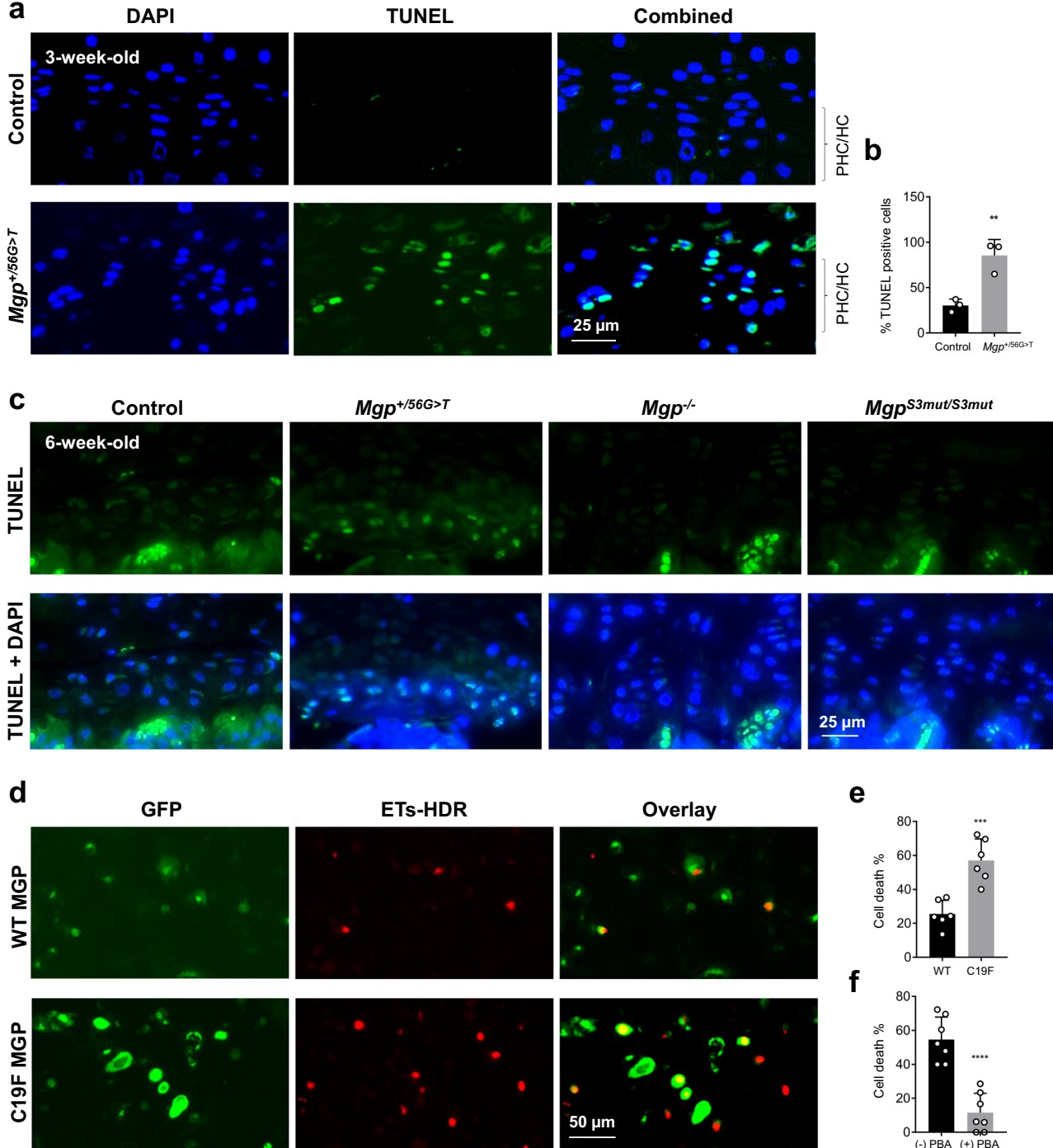

**Fig. 8 | The C19F variant in MGP increased cell death in the growth plate chondrocytes and in chondrogenic ATDC5 cells. a** TUNEL assay shows a drastic increase in the number of apoptotic chondrocytes in the vertebral sections of 3-week-old *Mgp⁺/⁵⁶ᴳ>ᵀ* mice compared to the control mice. **b**. Bar graphs showing the quantification of TUNEL positive cells (*n* = 3 individual image fields/group). **c** TUNEL assay on 6-week-old vertebral sections of the control, *Mgp⁺/⁵⁶ᴳ>ᵀ*, *Mgp⁻/⁻* and *Mgp^S3mut/S3mut* mice shows the clustered presence of apoptotic chondrocytes in the growth plates of *Mgp⁺/⁵⁶ᴳ>ᵀ* but not in other genotypes. **d** Ethidium homodimer-based cell death assay performed on ATDC5 cells transfected with *pMgp-GFP* or *pC19FMgp-GFP* vectors. Cells expressing C19FMGP-GFP fusion protein show increased deaths as evident by both GFP and ethidium homodimer staining (green

and red, respectively). **e** Quantification of the dead cells in transfected ATDC5 cells. Cells with both GFP and ethidium homodimer staining were counted and normalized by the total number of GFP-positive cells. Values were represented as percent of cell deaths (*n* = 6 individual image fields/group). **f** Treatment of ATDC5 cells expressing C19F MGP-GFP fusion protein with 4-phenyl butyric acid (4-PBA) prevented cell deaths. ATDC5 cells were transfected with the *pC19FMgp-GFP* plasmid and cultured in α MEM complete media with or without 5 mM 4-PBA for 48 h. The percent of cell deaths were calculated as described in (**e**) (*n* = 7 individual image fields/group). Error bars represent standard deviations. Data presented as mean ± SD. **$P < 0.01$; ***$P < 0.001$; ****$P < 0.0001$ vs. control (or WT) by two tailed unpaired *t* test. Actual *P* values and source data are provided as a Source Data file.

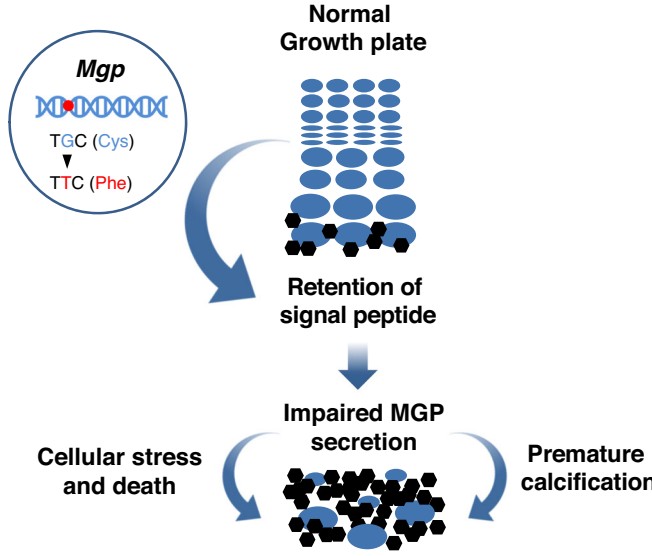

**Fig. 9 | A model describing the effects of C19F MGP expression on the growth plate.** The C19F variant in MGP results in the impaired processing of the signal peptide and the retention of the mutated protein in the ER. This in turn leads to ER stress and markedly increased deaths of the growth plate prehypertrophic/hypertrophic cells which are known to express MGP at high levels. The cellular alterations may affect the expression and secretion of WT MGP causing the premature closure of the growth plates and the associated skeletal anomalies.

affected individual from this report (Individual 4). As premature closure of growth plates was observed in both an affected individual and in our $Mgp^{+/56G>T}$ mice, this clinical finding is likely part of the phenotype associated with SED, $MGP$ type. The presence of thinner and prematurely mineralized growth plates could be explained by possible impairment of MGP's anti-mineralization functions as well as markedly increased apoptosis of the growth plate chondrocytes. While chondrocyte apoptosis appears to be the primary cause, other yet unknown cellular alterations may also contribute to this pathology. Collectively, the observed growth plate pathologies in $Mgp^{+/56G>T}$ mice appear to be caused by complex aetiologies which might involve a combined dominant negative as well as gain of function effects of the C19F variant.

We performed a thorough analysis of the bone remodelling parameters in our $Mgp^{+/56G>T}$ mice considering their severely reduced bone mass. Of note, bone mass was only evaluated in Individual 1 from this report in her adolescent years and was reported normal later. Increased incidence of fractures or abnormal healing following orthopedic surgeries were not observed in any affected individuals in this report. While more clinical data is needed to determine the effects of C19 variants on human bone mass, in the mutant mice both arms of bone remodelling—bone resorption by osteoclasts and bone formation by osteoblasts—were found to be affected. Since osteoclast numbers were lower in the mutant mice in comparison to the control mice and bone formation rate was not significantly affected, we concluded that the overall reduction of the osteoblast number is the primary cause of the poor bone mass in these mice.

While the bone mass is invariably reduced in $Mgp^{+/56G>T}$ mice, a key question remains open—what is the cause of the low osteoblast numbers in these mice that results in the low bone mass? This question can be answered, at least in part, by revisiting the theory that a subset of hypertrophic chondrocytes transdifferentiates to osteoblasts. Indeed, recent use of the in vivo cell-tagging techniques in mice confirmed that a substantial portion of hypertrophic chondrocytes transdifferentiate

to osteoblasts[23,24]. Based on these findings, we believe that the markedly increased apoptosis of prehypertrophic/hypertrophic chondrocytes in the growth plates is likely causing the reduction of osteoblast numbers in our mutant mice. At this stage, it is not clear whether there are yet unknown systemic or local effects of the 56G>T variant on the proliferation and differentiation of osteoblasts. However, considering that $Mgp$ is not expressed in osteoblasts in the endochondral bones[12], a local effect is unlikely.

$Mgp^{+/56G>T}$ mice displayed the major clinical features seen in the affected individuals in this report, namely the short stature, midface retrusion, platyspondyly, tracheal calcification as well as growth plate and intervertebral joint anomalies. As seen in the affected individuals, the mutant mice did not show any signs of vascular calcification, a feature that is present in MGP-deficient mice and in some individuals with KS. Unlike the affected individuals, however, the mutant mice died prematurely. This discrepancy in the life expectancy of humans with the C19F or C19Y variant and $Mgp^{+/56G>T}$ mice is not unusual. For example, MGP-deficient mice die within two months of age while most humans with KS lacking MGP have a normal or slightly reduced life expectancy compared to the general population[11]. MGP-deficient mice develop rapid arterial calcification leading to the rupture of the thoracic aorta which has been identified as the primary cause of their deaths[5]. However, since no vascular calcification has been detected in our $Mgp^{+/56G>T}$ mice, we can rule out this as the potential cause of their early lethality.

MGP deficiency has been shown to upregulate BMP signaling and arteriovenous malformation in mice[25]. Similar or yet unknown alterations of signaling events and associated pathologies affecting the MGP expressing tissues may lead to early lethality in $Mgp^{+/56G>T}$ mice. Alternatively, it is possible that severe bone loss resulting in poor energy metabolism together with the acute craniofacial phenotype in the mutant mice lead to early lethality. Although at present the precise cause of premature deaths of $Mgp^{+/56G>T}$ mice is unknown, its recapitulation of the major human traits caused by the C19F variant suggests its usefulness in mechanistic studies aimed to understand the human disease.

Our cell culture experiments were instrumental to uncover the mechanisms underlying the observed skeletal anomalies in mice. We initially performed transfection experiments with ATDC5 chondrogenic cells and demonstrated that the C19F variant in MGP results in impaired processing of the signal peptide and intracellular accumulation of the mutant protein resulting in its poor secretion. We used HEK-293 cells in the further analyses for two reasons—firstly, it is a human cell line thus it is suitable for testing the effects of a variant found in humans; and secondly, these cells are easily transfected resulting in high level production of plasmid-coded proteins. These attributes of HEK-293 cells are convenient for the downstream analyses of the proteins expressed by the transfected plasmids. It was reassuring to observe that the intracellular accumulation of the mutant protein and cell death patterns were similar in both ATDC5 and HEK-293 cell lines.

Our cell culture data shows that a portion of the mutant MGP appears to be processed correctly—i.e., cleaved at the end of the signal peptide. We believe that this is because the C19F mutation does not fully prevent the cleavage of the variant signal peptide, rather results in its inefficient cleavage and intracellular accumulation of the unprocessed protein. This explanation was supported by mass spectroscopic analyses as peptides both lacking and carrying a part of the signal peptide were found in the samples from cultures expressing the mutant protein, while WT cultures did not generate any peptide with any amino acids from the signal peptide.

For the current study, we constructed several plasmid vectors expressing native or mutated MGP with a FLAG[26] or GFP tag at the C terminal end; or expressed MGP and GFP separately from a single transcript. Use of these vectors not only helped us to bypass the

problem of not having a reliable antibody against mouse MGP, but also provided critical information on the effects of accumulated C19F MGP on cell behavior. Using these expression vectors in transfections experiments, we were able to demonstrate that mutant MGP when co-expressed with WT MGP may promote the intracellular retention of the latter, which may in turn result in a dominant negative effect. However, these findings should be interpreted with caution as they will need to be validated in vivo in future.

Abnormal accumulation of unprocessed proteins has been shown to cause ER stress, which can be detected by the upregulation of several well-established marker proteins. The increased presence of calnexin, an ER chaperone protein[27,28], in the C19F MGP expressing HEK-293 cells suggested an induction of unfolded protein response. Further, increased expression of CHOP, a marker for ER stress and activated intrinsic pathways of apoptosis[29], provided clues to the final cell fate upon intracellular accumulation of C19F MGP. In agreement with our analyses of unfolded protein response and ER stress markers, we observed markedly increased apoptosis of HEK-293 cells expressing the mutant proteins.

A previous study reported calnexin expression in the cartilaginous tissues of WT mice[30]. In agreement with this, we found expression of this chaperone in the prehypertrophic/hypertrophic zone of the growth plates in our control mice which was increased in the mutant mice. The increased expression of calnexin in the prehypertrophic/hypertrophic chondrocytes and remarkably higher apoptosis of these cells in the mutant mice show the complementarity of our in vitro and in vivo data. Furthermore, we demonstrated that two downstream mediators of ATF6 and PERK, including a common downstream response protein CHOP were upregulated in the growth plates of $Mgp^{+/56G>T}$ mice. Collectively, our in vitro and in vivo analyses suggest that ER stress-associated anomalies including apoptosis are likely the primary cause of the skeletal traits in the mutant mice. In addition, the induction of ER stress and abnormally high apoptosis of mutant MGP producing prehypertrophic/hypertrophic cells might have led to a reduction of extracellular MGP levels in the narrowed prehypertrophic/hypertrophic zones causing excessive mineral accumulation and premature closure of the mutant growth plates.

ER stress-induced apoptosis has been demonstrated to be a common mechanism leading to various skeletal disorders, such as metaphyseal chondrodysplasia, Schmid type (OMIM#156500), pseudoachondroplasia (OMIM#177170), and COL1A1-related osteogenesis imperfecta (OMIM#166200)[31–33]. The affected pathway(s) were targeted to study therapeutic strategies to improve or slow the clinical evolution of these progressive disorders. For example, carbamazepine was shown to reduce ER stress, increase bone growth rates and improve the skeletal phenotype in a mouse model with metaphyseal chondrodysplasia, Schmid type [32]. Also, treatment by 4-PBA, an FDA-approved pharmacologic agent to prevent ER stress might prevent or at least delay the progression of these skeletal phenotypes[34–36]. In line with these reports, our experiment showing reduced cell deaths in 4-PBA-treated ATDC5 chondrogenic cells expressing C19F MGP is very promising and potential therapeutic benefits are worth investigating further in animals and in humans. Other promising drugs that might be examined in this regard are caspase inhibitors. Indeed, several studies over the past years have indicated that caspases, enzymes regulating apoptosis, are attractive targets for therapeutic intervention in a wide range of diseases characterized by excessive apoptosis[37,38]. Future animal studies may provide insights on the yet unidentified pathways contributing to the skeletal traits caused by the C19F variant of MGP enabling further examination of various therapeutic approaches for this and other related skeletal dysplasias in humans.

In summary, our findings support that heterozygous variants in *MGP* altering the Cys19 residue cause an autosomal dominant spondyloepiphyseal dysplasia, which we suggest naming SED, *MGP* type. We show that the C19F variant in MGP leads to impaired processing of the signal peptide, which in turn leads to poor secretion and intracellular retention of the protein. We provide compelling evidence that the retained protein accumulates in the ER causing cellular stress and markedly increased apoptosis in chondrocytes, particularly in the prehypertrophic/hypertrophic zone of the growth plates. While our work does not examine additional mechanisms, such as alterations of signaling events regulating the differentiation of hypertrophic chondrocytes, it appears that the early loss of the matured chondrocytes, narrowing and premature calcification of the growth plates result in the shortening of the endochondral bones. We believe that the abnormal loss of prehypertrophic/hypertrophic chondrocyte pools destined to transdifferentiate to osteoblasts reduces the overall number of functional osteoblasts, which is likely the primary cause of the low bone mass phenotype in the mutant mice.

The current study has provided insight into the mechanisms underlying the skeletal anomalies caused by the Cys19 variants in MGP and demonstrated that this represents a disorder distinct from KS, both clinically and molecularly. The identified pathways affected by these variants can potentially be targeted to treat and manage SED, *MGP* type in the future.

## Methods

### Study approval
The clinical study was approved by the Children's Hospital of Eastern Ontario Research Ethics Board. Informed consent was obtained from all participants including the four affected individuals to participate in this study and they agreed to have their clinical data (medical information, photographs and genetic data) included in this publication. All mouse experimental procedures were approved by the animal care committee of McGill University and conducted in accordance with the IACUC's guidelines.

### Exome sequencing
Exome sequencing of the three affected and two unaffected individuals from Family 1 was performed in 2018 as part of the Care4Rare Canada research program[39]. Targeted exon capture was performed using the Agilent SureSelect Human All Exon 50 Mb (v5) enrichment kit and sequenced on an Illumina HiSeq 2000 using 2 × 100 bp chemistry. Alignment, variant calling, and annotation were done with a bcbio-based pipeline (https://github.com/bcbio/bcbio-nextgen), GEMINI, methods accumulated in FORGE and Care4Rare Canada Projects and custom annotation scripts (https://github.com/naumenko-sa/cre)[40–42]. Exome sequencing data was filtered for rare variants with high sequencing quality and with minor allele frequencies (MAF) < 0.001 in gnomAD v2.1.1 (https://gnomad.broadinstitute.org). Heterozygous and X-linked variants seen <7 times or homozygous and compound heterozygous variants seen <15 times out of 5220 alleles in the Care4Rare in-house call-set were selected. Two genomic analysts independently reviewed the list of rare variants and, given the suspected autosomal dominant inheritance in Family 1, prioritized candidate heterozygous variants shared between the three affected individuals, but variants following an X-linked or autosomal recessive inheritance were also analyzed. Forty-nine rare (MAF < 0.001 in gnomAD v2.1.1 and <7 allele counts in the Care4Rare call-set) heterozygous variants were shared by the three affected individuals from Family 1 and not present in the unaffected father and brother. Variants were sorted and prioritized using in silico prediction scores, phenotypic data associated with the gene in the OMIM database, gene function, gene constraint scores (https://gnomad.broadinstitute.org), genotype quality scores, and allele frequencies in the Care4Rare call-set and in the gnomAD database (https://gnomad.broadinstitute.org). The c.56G>T variant in *MGP* was selected as a strong candidate, and no other variants in known or novel genes have been retained as plausible candidates after exome analysis. Segregation of the *MGP* variant in all individuals from Family 1 (the two affected siblings, the unaffected father and brother, and the

affected mother) was confirmed by Sanger sequencing (primers available on request) at the Children's Hospital of Eastern Ontario (Ottawa, ON, Canada).

Exome sequencing of Individual 4 from Family 2 and her parents was performed on a clinical basis in a commercial laboratory (GeneDx, Maryland, USA).

Web resources:

gnomAD, https://gnomad.broadinstitute.org/gene/ENSG00000 111341?dataset=gnomad_r2_1.

GTEx, https://gtexportal.org/home/gene/MGP.

Matchmaker Exchange, https://www.matchmakerexchange.org/.

OMIM, https://www.omim.org/entry/154870?search=mgp&highlight=mgp.

CADD, https://cadd.gs.washington.edu/snv.

ClinVar, https://www.ncbi.nlm.nih.gov/clinvar/variation/432101/.

## Animal models

To generate $Mgp^{+/56G>T}$ mice, guide RNAs (gRNAs) close to the Cys19 residues were designed using the online software CRISPOR. Three different gRNAs were ordered from Synthego and targeting efficiency (ability to introduce insertions/deletions at the $Mgp$ loci) of each of them was determined by in vitro culture of gRNA-Cas9-injected blastocysts and subsequent DNA sequencing. Next, microinjection of the validated gRNA-Cas9 protein and repair ssODN template (carrying the desired TGC to TTC mutation) complex was performed at the Transgenic Core Facility, Goodman Cancer Research Centre, McGill University. The extracted DNAs from tail biopsies were PCR-amplified using the following target-specific primers:5′-GGCAAGTTTAGTGC-CAAGCC-3′ and 5′-ACATGCGCTGGAATGACAATG-3′. The resulting PCR products were subjected to the Sanger sequencing at Génome Québec, Montreal, Canada. Data analysis was performed using DNA Strider or SnapGene software.

In addition to $Mgp^{+/56G>T}$ and control WT mice, two mouse models $Mgp^{-/-}$ and $Mgp^{S3mut/S3mut}$ mice were used. We obtained $Mgp^{-/-}$ mice from Dr. G Karsenty[5]. $Mgp^{S3mut/S3mut}$ mice lacking the N-terminal serine residues were generated and reported recently by our lab[43]. The mouse lines were maintained at the vivarium of Shriners Hospital for Children, Canada following an animal use protocol approved by the animal care committee of McGill University. Mice were housed within a temperature-controlled room (21–22 °C) under a 12 h light/dark cycle and allowed free access to food and water.

## Radiography and X-ray micro-computed tomography (micro-CT) analyses

Both radiographic and micro-CT analyses of the mice were performed in the Center for Bone and Periodontal Research at McGill University. X-ray images were obtained with an XPERT-80 radiography imaging system (Kubtec Technologies Inc., Milford, CT). The samples were scanned at X-ray source power of 22 kV/600 µA. Micro-CT images of the skeletal samples were generated with a SkyScan 1172 micro-CT instrument (Skyscan, Bruker, Kontich, Belgium). This scanner is equipped with a sealed microfocus X-ray tube, with operating power ranging from 20 to 100 kV (0–250 mA), and X-ray charge-coupled device (CCD) camera with a cooled 4000 × 2096 pixels 12-bit CCD sensor. Briefly, the samples were scanned using an 0.5 Al filter with a spatial resolution of 12 µm at a voltage of 59 kV and a power of 10 W. The rotation was set at 0.4°/step for 180°. The exposure time was set at 3564 ms. Skyscan softwares were used for cross sectional reconstructions with NRecon, 3D reconstruction with CTAn and CTVol.

## Histology and histomorphometric analyses

For calcein double labeling, 5-week-old mice were injected intraperitoneally with 10 µl/g body weight of calcein solution (0.25% calcein and 1% NaHCO3 dissolved in 0.15 M NaCl) twice at a 3-day interval. Mice were euthanized 4 days after the second injection and skeletal tissues were processed for histomorphometric analyses. For plastic sectioning, vertebrae and long bones were fixed overnight in 4% paraformaldehyde/phosphate-buffered saline (PBS), embedded in methyl methacrylate, and sectioned (7-µm thickness). Von Kossa and van Gieson (VKVG), Von Kossa and Safranin O (VKSO), Toluidine blue or TRAP staining were applied. Stained bone sections were analyzed for bone volume/tissue volume (BV/TV), osteoblast count, osteoclast count, MAR and BFR/BS analyses using the Osteomeasure software (Osteometrics, Inc.). The region of interest (ROI) was selected in the lumbar vertebrae (L3 and L4) sections avoiding the areas adjacent to the cortical bones and the growth plates. Images were taken at room temperature using a light microscope (DM200; Leica) with a 5× (numerical aperture of 0.11), 20× (numerical aperture of 0.40) or 40× (numerical aperture of 0.65) objective. All histological images were captured using a camera (DP72; Olympus), acquired with DP2-BSW software (XV3.0; Olympus Canada Inc), and processed using Photoshop (Adobe).

## Generation of plasmid vectors

Plasmids $pMgp$-FLAG-IRES-GFP or $pC19FMgp$-FLAG-IRES-GFP used in cell culture experiments were generated using the parent vector $pIRES$-hrGFP (Agilent). MGP-FLAG or C19F MGP-FLAG proteins expressed by these vectors are coded by the WT or mutated $Mgp$ gene sequence, respectively with "in frame" FLAG tag coding sequence inserted immediately before the stop codon. Plasmids $pMgp$-GFP or $pC19FMgp$-GFP were constructed using the parent vector $pEGFP$ (Clontech). The open reading frame of WT or mutant $Mgp$ cDNA was subcloned "in frame" with the EGFP coding sequence to express the respective fusion proteins. Plasmid $pLVX$-IRES-Puro (Clontech) was used to generate $pMgp$-FLAG or $pC19FMgp$-FLAG constructs. The WT or mutated $Mgp$ gene sequences carrying the "in frame" FLAG tag coding sequence at the 3′ end expresses the respective tagged proteins. For all three sets of plasmids described above the WT MGP coding plasmids were constructed first. They were then used to mutate the Cys19 codon TGC to phenylalanine codon TTC by PCR mutagenesis (Q5® Site-Directed Mutagenesis Kit, New England Biolabs) using the following primers:

5′-GCAACCCTGTTCTACGGTGAGAAACCTCTC-3′ and 5′-CACGGC CAGCGCAGCCAG-3′.

## Cell culture and DNA transfection

ATDC5 cells (Sigma Aldrich #99072806, gift from Dr. Moffatt's lab at Shriners Hospitals for Children, Canada) and human embryonic kidney cells (HEK-293) (American Type Culture Collection CRL-1573, gift from Dr. St-Arnaud's lab at Shriners Hospitals for Children, Canada) were cultured in Gibco™ alpha MEM (Invitrogen #12571071) or Dulbecco's Modified Eagle's Medium (DMEM) (Winsent, 319-005-CL) supplemented with 10% fetal bovine serum (FBS, HyClone Laboratories, SH3039603) and 100 U/ml of penicillin–streptomycin at 37 °C under 5% $CO_2$ in a humidified incubator. Cells were plated and cultured in triplicates on four-well plates or 10 cm dishes until reaching 70% confluency. Then ATDC5 cells were transfected with the WT and C19F expression vectors $pMgp$-FLAG-IRES-GFP and $pC19FMgp$-FLAG-IRES-GFP or $pMgp$-GFP and $pC19FMgp$-GFP using Lipofectamine 3000 reagent (Fisher Scientific, L3000015). HEK-293 cells were transfected with the native or mutated $Mgp$ plasmid $pMgp$-FLAG or $pC19FMgp$-FLAG with or without co-transfection of $pMaxGFP$, a GFP expressing plasmid, by using Lipofectamine 3000. Upon 48 h (after transfection), cells were investigated for cell transfection efficiency by GFP signals under EVOS FL cell imaging system (Life Technologies); or examined by Western blotting, IF (Immunofluorescence), or cell death analysis.

## Gene expression analysis

Gene expression analysis was performed using a quantitative real-time PCR (qRT-PCR) system (Model 7500, Applied Biosystems). Total RNA was extracted from HEK-293 cells with TRIZOL reagent (Invitrogen

#15596026) and subjected to DNase I (New England Biolabs #M0303S) treatment. The first- strand cDNA synthesis and qRT-PCR were performed using a High-Capacity cDNA Reverse- transcription kit (Applied Biosystems #4368814) and Advanced qPCR master mix with SUPER-GREEN DYE (Wisent, 3800-435-UL), respectively. Relative gene expression was analyzed by SDS software (Applied Biosystems) using comparative CT and hypoxanthine guanine phosphoribo-syl transfer-ase (*Hprt*, a housekeeping gene) expression as an endogenous control. To calculate the delta cycle threshold (ΔCT) value, the mean CT value of the expression of a gene in a sample was first normalized to the mean CT value of *Hprt* expression in that sample. The ΔCT value of the calibrator sample was subtracted from that of the sample of interest to obtain the ΔΔCT value. The relative expression was reported as $2^{-\Delta\Delta CT}$. The data presented were obtained from 3 individual samples. The following primer sequences were used to detect *Mgp* expression: 5′-ACCCGAGACACCATGAAGAG-3′ and 5′-AGGACTCCATGCTTTCGTGA-3′.

## Immunoprecipitation using M2 FLAG beads

Total protein extracts from 48 h transfected ATDC5 cell lysates were added into radioimmunoprecipitation assay buffer containing 1 mM phenylmethylsulfonyl fluoride, 2 mM NaF, 0.5 mM $Na_3VO_4$ and 2 μg/ml leupeptin. The conditioned media of transfected ATDC5 and HEK-293 cells were harvested and centrifuged at 4 °C to remove dead cells and debris (15 min at $4000 \times g$). Next, the media (10 ml) were loaded onto Amicon Ultra-15 Centrifugal Filter Units (MWCO = 3 kDa, Millipore Sigma, UFC9003) and concentrated to ≤1 mL by repeated centrifugation at $4000 \times g$ at 4 °C. Conditioned media were then incubated with pre-washed anti-FLAG® M2 magnetic beads (Sigma-Aldrich, M8823-1ML) overnight at 4 °C. The beads were washed three times with wash buffer (1× PBS + 0.002% TWEEN 20), and the immunoprecipitates were denatured by incubating with the Laemmli buffer for 7 min at 100 °C and then subjected to SDS/PAGE.

## Western blotting

Total cell lysates and immunoprecipitates were resolved on 15% tris-glycine SDS-PAGE gel, transferred onto a polyvinylidene difluoride (PVDF) membrane (Bio-Rad, 1620177) and were detected using an anti-DYKDDDDK Tag (D6W5B) rabbit mAb (binds to same epitope as Sigma's Anti-FLAG® M2 Antibody) (1:1000 Cell Signaling Technology, 14793S), and 1:5000 dilution anti-rabbit HRP conjugated IgG as the secondary antibody (Cell Signaling Technology, 7074).

## Proteomics analysis

For each sample, a single stacking gel band containing all proteins was reduced with DTT, alkylated with iodoacetic acid and digested with trypsin. Extracted peptides were re-solubilized in 0.1% aqueous formic acid and loaded onto a Thermo Acclaim PepMap™ precolumn (ThermoFisher, 75 μM ID X 2 cm C18 3uM beads) and then onto an Acclaim™ PepMap™ EASY-Spray (ThermoFisher, 75 μM X 15 cm with 2 μM C18 beads) analytical column separation, using a Dionex Ultimate 3000 uHPLC at 250 nl/min with a gradient of 2–35% organic (0.1% formic acid in acetonitrile) over 2 h. Peptides were analyzed using a Thermo Orbitrap Fusion mass spectrometer operating at 120,000 resolution for MS1 with HCD sequencing at top speed (15,000 resolution) for all peptides with a charge of 2+ or greater. The raw data were converted to mgf format (Mascot generic format) for searching using the Mascot 2.5.1 search engine (Matrix Science) against human protein sequences (Uniprot 2020) or a protein FASTA database containing the wildtype and mutated forms of human MGP. The database search results were loaded onto Scaffold Q+ Scaffold_4.4.8 (Proteome Sciences) for statistical treatment and data visualization. Pinnacle (Optys Tech) was used to quantify all detected peptides using a MS1 quantification workflow (Targeted Quantification: Label Free DDA) wherein the peptide specific XICs from the raw mass spec data (.raw) were used to directly compare all identified peptide (*.dat) amounts across all experiments using precursor ion integrals (in counts). The mass spectrometry proteomics data have been deposited to the ProteomeXchange Consortium via the PRIDE partner repository (dataset identifier PXD043374). URL:https://www.ebi.ac.uk/pride/.

## Immunofluorescence and immunohistochemistry analyses

Cultured cell layers on chamber slides were washed and fixed with 4% paraformaldehyde in phosphate-buffered saline (PBS) 48 h after transfection. Cells were then permeabilized for 5 min using 0.25% Triton X-100 and then blocked in PBS with 2% BSA for 1 h at room temperature, followed by staining with primary antibody overnight at 4 °C. Cells were washed three times with PBS and incubated with the secondary antibody for 1 h at room temperature. Nuclei were labelled with DAPI (Sigma Aldrich, H33258). After washing 3× in 0.25% Triton X-100 buffer, slides were mounted by coverslips using Aqua-Mount medium (Thermo Scientific, TA-125-AM) and analyzed by the fluorescence microscopy using an EVOS FL cell imaging system (Life Technologies). The following primary antibodies were used: DYKDDDDK Tag (D6W5B) rabbit anti-FLAG monoclonal antibody (mAb) (1:800 Cell Signaling Technology, 14793S), ANTI-FLAG® M2 mouse mAb (1:800 Sigma-Aldrich, F1804), CHOP (L63F7) mouse mAb (1:400 Cell Signaling, 2895S) and calnexin (C5C9) rabbit mAb (1:400 Cell Signaling, 2679S). For secondary antibodies, we used Cy3 conjugated donkey anti-rabbit antibody (1:500 Jackson ImmunoResearch, 711-165-152) and Alexa Fluor 594 conjugated anti-mouse antibody (1:400 Cell Signaling, 8890S) for red fluorescence; while DyLight 488 conjugated anti-rabbit antibody (1:500 Jackson ImmunoResearch, 711-486-152), Alexa Fluor 488 conjugated anti-rabbit antibody (1:500 Jackson ImmunoResearch 711-545-152) and Alexa Fluor 488 conjugated anti-mouse antibody (1:500 Cell Signaling, 4408S) were used for green fluorescence.

For immunofluorescence of mouse tissues, decalcified lumbar vertebrae (in 15% EDTA; Fisher Scientific, BP118-500) from 3-week-old control and $Mgp^{+/56G>T}$ mice were embedded in paraffin and sectioned (5 μm thickness). Sections were deparaffinized, processed for antigen retrieval, and blocked with 5% bovine serum albumin (BSA; Fisher) in Tris-buffered saline (TBS)–0.25% Triton X-100 (TBST) for 30 min at room temperature in a humidified chamber. The sections were then incubated overnight at 4 °C in a humidified chamber with the primary antibody. Slides were washed three times with TBST and incubated with the secondary antibody for 1 h. After the DAPI staining followed by three washes with TBST, slides were mounted as above. Images were captured by the EVOS FL cell imaging system (Life Technologies) or a confocal microscope (Zeiss) at the imaging platform of McGill University Health Center. The following primary antibodies were used: anti-Collagen II (1:400 abcam, ab21291), anti-Aggrecan antibody (1:50 abcam, ab36861), anti-Collagen X (1:500 abcam, ab260040), calnexin (C5C9) rabbit mAb (1:400 Cell Signaling 2679S), anti-ATF6 antibody (1:100 abcam, ab37149), phospho-eIF2α (Ser51) (D9G8) XP® rabbit mAb (1:100 Cell Signaling, 3398), XBP-1s (D2C1F) rabbit mAb (1:100 Cell Signaling, 12782) and CHOP (L63F7) mouse mAb (1:400 Cell Signaling, 2895). The secondary antibodies used were Cy3 conjugated donkey anti-rabbit antibody (1:500 Jackson ImmunoResearch, 711-165-152), DyLight 488 conjugated donkey anti-rabbit IgG (1:1000 Jackson ImmunoResearch, 711-486-152) or Alexa Fluor 488 conjugated anti-mouse antibody (1:500 Cell Signaling, 4408S).

## Detection of cell death

To identify the dead cells in transfected cultures, cells were rinsed with PBS twice, then incubated in alpha MEM (ATDC5 cells) or DMEM (HEK-293 cells) medium containing 2 mM ethidium-homodimer (EthD-1; Invitrogen, E1169) for 40 min at 37 °C. Next, cells were rinsed in PBS three times, nuclei were labeled with DAPI (for HEK-293 cultures only)

and cover slips were mounted on slides using Aqua-Mount medium (Thermo Scientific). Images were captured by fluorescence microscopy using an Evos FL imaging system (Life Technologies).

To examine the effects of ER stress inhibition on the cell survival, ATDC5 cells were transfected with *pC19F Mgp-GFP* plasmids and cultured in alpha MEM medium supplemented with or without 5 mM 4-Phenylbutyric acid (4-PBA) (Sigma, P21005) for 48 h. Cells were then incubated with 2 mM ethidium homodimer for assessment of cell death.

TUNEL assays were performed on both fixed HEK-293 cells and tissue sections using a DeadEnd Fluorometric TUNEL System (Promega #G3250) following the instructions of the manufacturer. Lumbar vertebrae samples from 3-week-old mice were decalcified in 15% EDTA (Sigma-Aldrich) in PBS and embedded in paraffin to cut 5 μm sections. Plastic sections (7 μm) of vertebrae from 6-week-old mice were decalcified in 15% EDTA in PBS for 36 h at 4 °C. Upon enzymatic fluorescent labeling of fragmented DNA, nuclei were labeled with DAPI. TUNEL-positive cells were visualized using EVOS FL cell imaging system (Life Technologies) and quantified using ImageJ software (National Institutes of Health).

## Statistics and reproducibility

Bar graphs are shown as mean ± SD. Statistical analyses were performed by Student's *t* test or analysis of variance (Tukey's multiple-comparisons test) using GraphPad Prism software (*, $p < 0.05$; **, $p < 0.01$; ***, $p < 0.001$; ****, $p < 0.0001$). Representative micrographs were selected from at least three independent images from the replicates.

## Reporting summary

Further information on research design is available in the Nature Portfolio Reporting Summary linked to this article.

## Data availability

The c.56G>T variant in *MGP* was submitted to ClinVar (https://www.ncbi.nlm.nih.gov/clinvar/) (GenBank: NM_000900.3; accession number SCV004024152). Due to data privacy considerations and in line with Care4Rare Canada Consortium policy, the patient raw exome datasets supporting this study have been deposited in Genomics4RD[44], Canada's rare disease genomic data repository, and are available through controlled access by contacting the platform (https://www.genomics4rd.ca/). Raw exome data consented under Care4Rare Canada Consortium is available for secondary use for rare disease research only after execution of a data sharing agreement and requests are responded to within 2 weeks. The mass spectrometry proteomics data generated in this study (dataset identifier PXD043374) can be accessed at URL:https://www.ebi.ac.uk/pride/. All data supporting the findings described in this manuscript, excepting the raw exome data described above, are available in the article and its Supplementary Information files, and from the corresponding author upon request. Source data are provided with this paper.

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

## Acknowledgements

We dedicate this study to Dr. Ophélie Gourgas, who passed away on January 2, 2023, during the completion of this project. SA and AE con-tributed equally to this study. The authors would like to thank the patients and their families who participated in this study. We thank Dr. Philippe Campeau for discussion on the findings and Kyoungmi Bak for sample organization. This work was supported in part by the Care4Rare Canada Consortium funded by Genome Canada and the Ontario Genomics Institute (OGI-147), the Canadian Institutes of Health Research (CIHR), Ontario Research Fund, Genome Alberta, Genome British Columbia, Genome Quebec, and Children's Hospital of Eastern Ontario (CHEO) Foundation. This work was also supported in part by a catalyst grant to MM from the Canadian Rare Diseases Models and Mechanisms (RDMM) Network funded by the CIHR (RCN-137793 and RCN-160422) and European Research Project on Transnational Research (EJP RD-JTC-1-2019: 294028). O.G. was supported by the Réseau de Recherche en Santé Buccodentaire et Osseuse (RSBO). A.E. and G.L. were supported by Children's Hospital Academic Medical Organization (CHAMO) Clinical Fellowship Awards. G.L. was also supported by the Broad Institute of MIT and Harvard Center for Mendelian Genomics (grants UM1 HG008900, U01 HG0011755 and R01 HG009141). K.M.B. was supported by a CIHR Foundation Grant (FDN-154279) and a Tier 1 Canada Research Chair in Rare Disease Precision Health.

## Author contributions

Project development, experimental design, experimentation and data analysis: O.G., G.L., A.E., S.A., A.D., J.L., R.S.C., S.M., S.N. and M.M. Manuscript preparation: O.G., G.L., A.E., S.A., J.L., M.B.B., K.M.B., and M.M. Conceptualization and supervision of the project: K.M.B. and M.M. All authors have approved the final version of the article.

## Competing interests

The authors declare no competing interests.

## Additional information

## Care4Rare Canada Consortium

**Kym M. Boycott**[2,3]

A full list of members and their affiliations appears in the Supplementary Information.

