## [Peer Review File · Nature Communications]

Heterozygous variants in MGP lead to endoplasmic reticulum stress causing spondyloepiphyseal dysplasiaReviewers' comments:

Reviewer #1 (Remarks to the Author):

The paper by Gourgas addresses a novel heterozygous variant of matrix Gla protein (MGP), a protein when absent known to have deleterious effects in cartilage and vessels. The authors describe a Cys19 mutation causing spondyloepiphyseal dysplasia, a phenotype distinct from Keutel syndrome.

The authors describe 4 individuals from two families with Cys19 alterations, which they confirmed by Crispr/Cas9 insertion of the mutation in mice, showing the same phenotype as in men.

Comments:

Abstract: the authors refer to that MGP is a secreted protein, which is true considering MGP is post-translationally modified by phosphorylation and carboxylation. However, the precise role and function of MGP is not unraveled, with also an intracellular function of MGP that is possible. The authors need to correct/ adapt this and elaborate on this.

Two single nucleotide substitutions in the MGP gene alter the conserved Cys19 residue. What does this do with the conformation (and indirectly function) of MGP? The authors need to provide data if one of the posttranslational modifications is affected.

Further the authors show ER accumulation of mutant protein in vivo, detected by IHC. The confirm this in HEK cells that also in vitro cellular ER stress is taken place. Why did the authors use HEK cells, as it is known that MGP is predominantly expressed by chondrocytes (and smooth muscle cells)? The authors further measure Calnexin and CHOP as markers of ER stress. CHOP is only one of three pathways of the unfolded protein response. More general markers downstream must be included, such as GRP78 and/or GRP94.

What is the life expectancy of humans with the Cyst19 mutation, as mice die within 2 month. And why is this discrepancy as one of the four affected patients was 52 years of age at time of inclusion.

The authors need to show (data/ results) of why mice die at this young age, and of what cause.

Warfarin results in impairment of MGP carboxylation, as well as to ER stress (Furmanik et al., ATVB 2021). This indicates that the C19 mutation is not unique in causing ER stress via MGP. The authors need to show how the Cys19 mutation affects ER stress, and what the downstream features of Cys19 (in relation to warfarin treatment) are as this indicates a broader clinical implication of the findings.

Reviewer #2 (Remarks to the Author):

This is a very interesting and well written paper describing heterozygous mutations in MGP causing a novel form of chondrodysplasia. Generally, these data support the conclusions of the authors. Nevertheless there are some weaknesses to the study which will need addressing. Some of the images are not good at all.

I'm not qualified to comment on the clinical and radiological findings, although they are consistent with a form of short trunk dwarfism, which they propose to call SED, MGP type, which seems reasonable to me.

The authors use NGS to identify novel variants in two families, both of which affect a conserved cysteine residue. Their pipeline appears robust, but it might have been helpful for full details of their sequencing to be in the supplemental e.g. how many variants identified in total.

Following identification of the putative disease-causing variant the authors then generated a 'knock-in' mouse model using Crispr/Cas9. The mouse clearing has a relevant phenotype and a very short. The authors should provide detail as to why the mice die early.

How have the authors determined that there was no 'off-target effect' in the rest of the mouse genome? Surely sequencing just MGP mRNA is not sufficient. Furthermore, there is no detail of the breeding strategy; it's my understanding that any 'off targets effects' should be bred off through outcrossing. Is it possible that lethality is due to an 'off-target effect'?

The authors undertake comprehensive bone phenotyping using radiographs and micro-CT, which clearly demonstrates the bone abnormalities. There are no scale bars on most of the images in Figure 3. Furthermore, are the images in Figure 4A the same size? A scale bar on the control figure would have been helpful.

The growth plate images in Figure 5 are very poor quality and are they matched? Why study 6-weeks and not for eg 3 weeks, which would give a much better image. Clearly the growth plate in mutant is very disrupted, but I am not convinced that the black spot is showing premature mineralisation – that easily could be an artefact from sectioning, indeed there a quite a few gaps in staining in the mineralised zone where it could have broken off from. A lot of the authors hypothesis depends on this image and therefore if this is real then it needs to be quantified – how many sections from how mice show premature mineralisation. Are the images in Fig 5 B actually matched – there is a complete lack of red staining in the mutant – is this correct? Would indicate there is no growth plate in the mutant.

The collagen X antibody is dreadful, which is there intracellular staining? Nothing can be interpreted from this experiment.

I have quite a few problems with Figure 6. I am surprised that the authors could not purchase a good MGP antibody, there are lots on the market and not being able use on to study the mouse model is a weakness of the study. To overcome this they expressed WT and mutant MGP in cells.

In figure 6D and E, what is the 35 kDa band? – I can't find any reference that MGP dimerises – so can it be an artefact? Furthermore, in Fig6D a large proportion of the mutant MGP appears to be processed correctly - i.e. correctly cleaved signal peptide to give the 15 kDa band – how do the authors reconcile this?

In addition, it is the unprocessed (17 kDa band) species that is secreted in the mutant cell model – please explain? Why is this 17 kDa band not detected by mass spec? The authors should also run these gels unreduced because the mutant MGP should have an additional unpaired cysteine, which may cause the mutant MGP to aggregate.

The authors hypothesis that mutant MGP is retained within the ER and causes stress – this is where a good MGP antibody and in vivo validation would have been ideal. The authors only look at CHOP and calnexin as ER-stress markers (Fig 7). They need to examine markers for all branches of the UPR – I would suggest BiP and also Xbp1 splicing, ATF6 cleavage and phosphorylation of eif2a.

The authors next attempted to validate ER-stress in the mouse model (Figure 8). Unfortunately, the calnexin antibody is not good enough – why all the extracellular staining? It's just not specific and no conclusions can be drawn from this experiment. Furthermore, why is DAPI staining the bone matrix, particularly in the mutant – that should not be happening. In the hypertrophic cells (shown by red arrows in mutant) the DAPI appears to be staining the lacunae occupied by the cell and not nucleus at all. This need to be repeated before any meaningful interpretation can be made. I'm not convinced by the EM image (there is also no wt control) those large areas don't look like distended ER to me – there are no ribosomes. In figure 8C I'm not convinced the images are correctly labelled – to me they show

rounded resting zone chondrocytes – there seems to be little evidence of stacked proliferative zone chondrocytes.

Reviewer #3 (Remarks to the Author):

Matrix Gla protein encoded by MGP is a secreted inhibitor of calcification in the extracellular matrix. MGP also negatively regulates BMP activity. Over expression of MGP decreases mineralization, delays chondrocyte maturation and inhibits endochondral ossification (PMID: 10579728) while loss of function causes abnormal excessive vascular calcification and premature bone mineralization, reduced bone growth and osteopenia (review PMID: 16612082). It is therefore an important regulator of physiological mineralization. Mutations in MGP cause Keutel syndrome (KS) an autosomal recessive disease affecting the skeleton such as hands, and limbs and midface malformation and in which there is abnormal increase in cartilage and arterial calcification. Other abnormalities occur, such as hearing loss, congenital heart defect. In this paper Gourgas et al describe 4 affected spondyloepiphyseal skeletal dysplasia individuals with heterozygous mutations in MGP that alter Cys19 to phe or tyr. This newly described spondyloepiphyseal dysplasia is interesting because it is an autosomal dominant mutation in MGP, of unknown etiology. In the paper the authors generate a mouse model that recapitulates the human phenotype. Because the mutation affects the signal peptide the authors hypothesise that the mutation will affect the secretion of MGP and trigger ER stress. They test this hypothesis in cells and the mouse model. The study is important because it could potentially shed new insights into the role of MGP, the potential dominant negative molecular mechanism underlying heterozygosity for the MGP Cys19 mutation. It also could extend the examples in which skeletal dysplasias are caused by the impact of ER stress on chondrocyte differentiation and bone formation. However while the study has potential interest, the major issue is that the depth of investigation is insufficient to define clearly the underlying molecular mechanism(s) and explain how heterozygosity for the MGP Cys19 affects chondrocyte differentiation and cause the bone defects as outlined below. The role of ER stress in the underlying molecular pathogenesis is incompletely established.

Major concerns

1. One major concern is that the characterization of the phenotype is not in sufficient depth. The study of the changes in humans and mice begin after the phenotype has manifested – the youngest patient is 2 years 4 months and at that age the epiphyses is already smaller; the earliest stage the Mgp 56G>T mice are analysed for skeletal defects e.g. bone lengths and bone mass is at 6 weeks – that is equivalent to a young human adult. The age of mice analysed in Fig. 4 and Fig. 8 are not stated. In Fig. 5, 3 week (equivalent to adolescent human) and 6 week mice are described. The stage and onset of the abnormal phenotype should be characterized especially since Mgp is already expressed in the developing cartilage at E14.5. In addition a prerequisite of the study must be characterization of the expression of Mgp 56G>T/MGPCys19. In which cells in the developing bone is the mutant expressed. Is it expressed in perichondrium/periosteum? In osteocytes? This has not been carried out.

2. The authors have not distinguished sufficiently between whether MGP Cys19 causes complete loss of function of MGP vs an impact of MGP Cys19 on wild-type MGP vs a neomorphic impact. That is, if or how the mutant affects the normal allele. Is the mutation effectively creating a null or has a dominant negative/new effect? In mice loss of MGP function results in increased calcification. In the MGP Cys19 patient 4 premature calcification is reported. Low bone mass and decreased mineralization is reported for Mgp 56G>T heterozygote mice. A comparison of the Mgp 56G>T heterozygote with the homozygous null would be informative. Further it would be important to compare the phenotypes of homozygous Mgp 56G>T and also compound Mgp 56G>T /null, to assess dominant interference with wild-type MGP .

3. Why is there less bone? The phenotype characterization of the chondrocyte differentiation defect and possible impact on osteoblast differentiation resulting in less bone, are superficial, with inadequate use of key molecular markers to pinpoint what phase of chondrocyte or osteoblast differentiation is affected. Using alcian blue staining (for proteoglycans in cartilage) is hardly sufficiently specific to determine whether late hypertrophic chondrocyte differentiation is affected. Indeed in Fig. 6 A and B only histological staining is used. In those sections the von kossa staining obscures assessment of the chondrocyte morphology. Little information can be surmised on any impact on osteoblast differentiation. Tibia and vertebra are shown in Fig 5 A and B, but the end-plate and growth plate of the IVD are not clearly shown. There also seems to be a difference in the morphology of the nucleus pulposus but that is not mentioned or investigated. Fig 5C shows type X collagen staining but it is of the 3 week vertebrae, and are these sections of the growth plate or endplate ? And what is the impact on both those structures as chondrocyte hypertrophy is involved in both. And the impact on the secondary ossification centre should be described too.

4. The authors then go on to test the hypothesis that inability to secrete MGP C19F triggers ER stress causing apoptosis of chondrocytes and possibly their differentiation to osteoblasts. However since there are 2 sources of osteoblasts – the periosteum and the hypertrophic chondrocytes it could well be an impact on periosteal osteoprogenitors as well. Given that many papers and reviews are available on the impact of ER stress on chondrocyte differentiation in vivo and in vitro (e.g. in ATDC5 cells) (e.g. PMID: 33307190, PMID: 34297411, PMID: 30902258 , PMID: 27002737) the authors should have included more convincing data and insights into a) which arm of the UPR/ER stress signaling is activated that could explain the chondrocyte differentiation defect . Furthermore give that several papers have been published showing the feasibility of overcoming at least partially the impact of the ER stress using pharmacological methods e.g. carbamazepine (CBZ), ISRIB and 4PBA (PMID: 32399188, PMID: 28920921, PMID: 30024379, PMID: 34990412), it would be important to demonstrate some rescue effects of either inhibiting the UPR or promoting proteostasis to reduce intracellular accumulation, at least in cell culture.

5. Overexpressing MGP in the developing limb has been shown to delay chondrocyte maturation and inhibit cartilage mineralization (PMID: 10579728). To what extent is the impact of MGP Cys19 due to increased stability/persistence of the protein versus triggering ER stress?

6. The authors have previously shown in Mgp null mice that there was increased apoptosis in chondrocytes. To what extent is the apoptosis triggered in MGP Cys19 due to loss of function versus or the activation of the UPR? In that regard the characterization of UPR markers in vivo in the mutants was rather superficial. Chop has multiple functions that can be anti-apoptotic or pro-apoptotic and can be activated by other factors not exclusively by the UPR. It would also have been more convincing if a chondrocyte progenitor cell line (e.g. ATDC5) has been used to demonstrate the accumulation of MGP

Cys19 in the ER and impact on differentiation in vitro, rather than kidney cells. Furthermore co-staining of the mutant protein with an ER marker would provide stronger evidence for accumulation in the ER.

7. Would the authors discuss what is the implication of a lack of aorta calcification in the Mgp 56G>T mice?

RESPONSE TO REVIEWERS' COMMENTS

We thank all the reviewers for their very constructive comments. We have addressed all the concerns raised by the reviewers either by performing additional experiments or by textual modifications in the revised manuscript. We believe that the overall quality of our study has improved significantly, and it is now suitable to be considered for publication in Nature Communications.

Reviewer's comments and a point-by-point response have been provided below. Please see the **Figures for the reviewers** at the end of this response.

Reviewers' comments:

Reviewer #1:

The paper by Gourgas addresses a novel heterozygous variant of matrix Gla protein (MGP), a protein when absent known to have deleterious effects in cartilage and vessels. The authors describe a Cys19 mutation causing spondyloepiphyseal dysplasia, a phenotype distinct from Keutel syndrome.

The authors describe 4 individuals from two families with Cys19 alterations, which they confirmed by Crispr/Cas9 insertion of the mutation in mice, showing the same phenotype as in men.

Thank you for reviewing this manuscript. Please find below our point-by-point response.

Comments:

1. Abstract: the authors refer to that MGP is a secreted protein, which is true considering MGP is post-translationally modified by phosphorylation and carboxylation. However, the precise role and function of MGP is not unraveled, with also an intracellular function of MGP that is possible. The authors need to correct/ adapt this and elaborate on this.

Answer: We thank the reviewer for this insightful comment. We have modified the sentence in the Abstract.

2. Two single nucleotide substitutions in the MGP gene alter the conserved Cys19 residue. What does this do with the conformation (and indirectly function) of MGP? The authors need to provide data if one of the posttranslational modifications is affected.

Answer: The reviewer has alluded to an interesting concept which might explain how the novel mutation may affect MGP's post-translation modifications and functions. To examine this, we have compared the growth plate phenotypes of the newly generated *Mgp*^{+/^{56G>T} mice (carrying the dominant C19F mutation in MGP) to that of MGP-null and recently reported *Mgp*^{S3mut/S3mut} mice which produce mutant MGP lacking the conserved serine residues which undergo posttranslational phosphorylation (shown to be critical for MGP's anti-mineralization function).}

Ref: <https://pubmed.ncbi.nlm.nih.gov/35418245/>). We now report here that the growth plate phenotypes in *Mgp*^{+/^{56G>T} mice markedly differ from that of MGP-null and *Mgp*^{S3mut/S3mut} mice. The narrowed fully calcified hypertrophic zone (**Fig. 5A,B**) and abnormally high number of apoptotic cells}

(**Fig. 10C**) are not present in the growth plates of MGP-null or *Mgp*^{S3mut/S3mut} mice. These findings suggest that the complete loss of MGP or lack of an essential post-translational modification in MGP is not the cause of the observed phenotype in the mice expressing C19F MGP.

3. Further the authors show ER accumulation of mutant protein in vivo, detected by IHC. They confirm this in HEK cells that also in vitro cellular ER stress is taken place. Why did the authors use HEK cells, as it is known that MGP is predominantly expressed by chondrocytes (and smooth muscle cells)?

Answer: Thank you for asking this important question. We used HEK293 cells for two reasons – firstly, it is a human cell line thus it is suitable for testing the effects of a mutation found in humans; and secondly, these cells are easily transfected resulting in high level production of exogenous proteins. These attributes of HEK293 cells are convenient for the downstream analyses performed on the expressed proteins. Nevertheless, following the reviewers' suggestions we performed new transfection experiments with ATDC5 chondrogenic cells transfected by the mutated *Mgp* construct and showed that a wild type MGP band and a higher molecular weight band corresponding to C19F MGP were both present in the cell extracts but not to the same extent in culture media by Western blotting experiments. These new findings are the same as in the case of HEK293 cells that we reported in the previous version (**Fig. 6C and D**). In addition, intracellular accumulation of the mutant protein (**Fig. 6E**) and resulted enhanced cell death (**Fig. 10D**) that we observed with the transfected HEK293 cells are all recapitulated in ATDC5 cells expressing C19F MGP.

4. The authors further measure Calnexin and CHOP as markers of ER stress. CHOP is only one of three pathways of the unfolded protein response. More general markers downstream must be included, such as GRP78 and/or GRP94.

Answer: We agree that our initial analyses of ER stress markers were rudimentary. We have now performed more thorough analyses of ER stress markers in the growth plates of control and mutant mice. Based on the published studies, we prepared a scheme (**Fig. 9B**) that depicts the three major pathways regulating CHOP expression. We have now included immunofluorescence images showing the expression and localization of downstream mediators of all three major arms regulating ER stress in the growth plates of control and *Mgp*^{+ / 56G>T} mice (**Fig. 9C, D and E**).

5. What is the life expectancy of humans with the Cyst19 mutation, as mice die within 2 months. And why is this discrepancy as one of the four affected patients was 52 years of age at time of inclusion.

Answer: At the moment, we cannot comment on the life expectancy of humans carrying the C19F mutation. The traits are more severe in mice, which is not unusual. For example, MGP-null mice (a model for Keutel syndrome) die within two months of age while humans with Keutel syndrome lacking MGP do not show unusually shorter life span.

(<https://www.frontiersin.org/articles/10.3389/fcell.2021.642136/full>).

6. The authors need to show (data/ results) of why mice die at this young age, and of what cause.

Answer: We thank the reviewer for asking this question. While MGP is highly expressed in the vascular and cartilaginous tissues, weak expression has been detected in multiple tissues. Since, we did not observe any vascular calcification (**Supplemental Fig. 4**; note MGP-null mice develop rapid vascular calcification and die within two months of age) or any other gross abnormalities of the vascular tissues, we can rule out the expression/accumulation of the mutated protein in the vascular tissues as a potential cause of early lethality. It is possible that severe bone loss resulting in poor energy metabolism together with the severe craniofacial phenotype in the mutant mice lead to early lethality. Indeed, it has been reported that severely low bone mass can lead to premature deaths (Bucay et al. 1988, <https://www.ncbi.nlm.nih.gov/pmc/articles/PMC316769/>). Unfortunately, a thorough analysis to identify the cause of death will require multiple new mouse models with conditional expression of the mutant gene. These experiments are beyond the scope of our current study which is focused on the understanding of the traits seen in the human patients.

7. Warfarin results in impairment of MGP carboxylation, as well as to ER stress (Furmanik et al., ATVB 2021). This indicates that the C19 mutation is not unique in causing ER stress via MGP. The authors need to show how the Cys19 mutation affects ER stress, and what the downstream features of Cys19 (in relation to warfarin treatment) are as this indicates a broader clinical implication of the findings.

Answer: We agree that other mechanisms to induce ER stress may exist. However, our recent comparisons of the growth plate phenotypes of the mutant mice expressing C19F MGP with that of MGP-null mice (see comment above and **Fig. 5A, B** and **Fig. 10C**) implies that the complete loss MGP function (and therefore the loss of its post-translational modifications) are not the reason why we see the observed traits in mice expressing C19F MGP.

Our in vitro and in vivo data convincingly show impaired processing and accumulation of mutant MGP protein in the ER (**Fig. 6, 7 and 8C**). This causes ER stress and abnormally high apoptosis of the hypertrophic chondrocytes which is not seen in whole body MGP-null mice (**Fig. 10C**). We have now provided a thorough characterization of the ER response pathways and show that prevention of ER stress by treating cells expressing the mutant protein with 4 phenyl butyric acid (4PBA) significantly reduces cell death (**Fig. 10D,E,F**). Additionally, we have examined the growth plates and long bone lengths of mice with chondrocyte-specific ablation of γ -glutamyl carboxylase deficient (GGCX; enzyme that converts the conserved Glu residues to Gla residues) mice (for another manuscript) and again the growth plate phenotype (not shown) and the long bone lengths (**Fig. 3 for Reviewers**; see at the end of this document) are not the same as seen in the *Mgp*^{+/*G56>T*} mice.

Reviewer #2:

This is a very interesting and well written paper describing heterozygous mutations in MGP causing a novel form of chondrodysplasia. Generally, these data support the conclusions of the authors. Nevertheless, there are some weaknesses to the study which will need addressing. Some of the images are not good at all. I'm not qualified to comment on the clinical and radiological findings, although they are consistent with a form of short trunk dwarfism, which they propose to call SED, MGP type, which seems reasonable to me.

Thank you for finding our paper interesting. We have added new data and modified the text to address the concerns. Please find below our point-by point response.

1. The authors use NGS to identify novel variants in two families, both of which affect a conserved cysteine residue. Their pipeline appears robust, but it might have been helpful for full details of their sequencing to be in the supplemental e.g. how many variants identified in total.

Answer: We provided more details on the exome analysis of affected individuals in the methods section of the manuscript.

2. Following identification of the putative disease-causing variant the authors then generated a 'knock-in' mouse model using Crispr/Cas9. The mouse clearly has a relevant phenotype and a very short. The authors should provide detail as to why the mice die early.

Answer: We thank the reviewer for asking this question. While MGP is highly expressed in the vascular and cartilaginous tissues, weak expression has been detected in multiple tissues. Since, we did not observe any vascular calcification (**Supplemental Fig. 4**; note MGP-null mice develop rapid vascular calcification and die within two months of age) or any other gross abnormalities of the vascular tissues, we can rule out the expression/accumulation of the mutated protein in the vascular tissues as a potential cause of early lethality. It is possible that severe bone loss resulting in poor energy metabolism together with the severe craniofacial phenotype in the mutant mice leads to early lethality. Indeed, it has been reported that severely low bone mass can lead to premature deaths (Bucay et al. 1988, <https://www.ncbi.nlm.nih.gov/pmc/articles/PMC316769/>). Unfortunately, a thorough analysis to identify the cause of death will require multiple new mouse models with conditional expression of the mutant gene. These experiments are beyond the scope of our current study which is focused on the understanding of the traits seen in the human patients.

3. How have the authors determined that there was no 'off-target effect' in the rest of the mouse genome? Surely sequencing just MGP mRNA is not sufficient. Furthermore, there is no detail of the breeding strategy; it's my understanding that any 'off targets effects' should be bred off through outcrossing. Is it possible that lethality is due to an 'off-target effect'?

Answer: We apologize that the previous version of the manuscript did not provide detailed information on the generation of the *Mgp*^{+/*56G>T*} mice. Actually, all the mutant mice were generated either by direct microinjections of SS-ODN, gRNAs and Cas9 protein complex (F0 generations) or by in vitro fertilization (IVF) using the sperm from a F0 male, which resulted in F1 or F2 generation mutant mice. The mutant mice generally die within 2 months of age and are unable to breed.

Since all the *Mgp*^{+/*56G>T*} mice generated from different microinjection and IVF events expressing C19F MGP showed the reported skeletal traits, we concluded that it is highly unlikely that they can be caused by the same off-target effects. We have now included a Table in **Fig. 2E** showing the number of mutant mice generated by microinjections and IVF and their full association with the phenotypes.

In addition, we followed the common practice to rule out the possible off target effects—we used an online application to examine whether there is any chromosomal sequence that might be targeted by the gRNA we used in our experiment. We found one possible genomic locus which can be targeted

by our gRNA. However, DNA sequencing of the PCR products amplified using locus-specific primers did not identify any alteration of the original sequence (**Supplemental Fig. 3**)

4. The authors undertake comprehensive bone phenotyping using radiographs and micro-CT, which clearly demonstrates the bone abnormalities. There are no scale bars on most of the images in Figure 3. Furthermore, are the images in Figure 4A the same size? A scale bar on the control figure would have been helpful.

Answer: Thank you for this important suggestion. The images were indeed taken at the same magnification. We have now included those changes in the revised version.

5. The growth plate images in Figure 5 are very poor quality and are they matched?

Answer: We have included better quality new images which are all matched.

6. Clearly the growth plate in mutant is very disrupted, but I am not convinced that the black spot is showing premature mineralisation – that easily could be an artefact from sectioning, indeed there are quite a few gaps in staining in the mineralised zone where it could have broken off from. A lot of the authors hypothesis depends on this image and therefore if this is real then it needs to be quantified – how many sections from how mice show premature mineralisation.

Answer: We apologize for wrongly presenting this image. A new image has been included (**Supplemental Fig. 5**). We confirm that the growth plate abnormalities are indeed the phenotype, not due to any kind of sectioning or staining artefacts (e.g. folding of the section). We have now included multiple growth plate images in a Figure for Reviewer (see at the end of this document) for better comparisons (Also see **Fig. 5**).

7. Are the images in Fig 5 B actually matched – there is a complete lack of red staining in the mutant – is this correct? Would indicate there is no growth plate in the mutant.

Answer: We confirm that the growth plate abnormalities are indeed the phenotype, not due to any kind of sectioning or staining artefacts as all different mice showed the same growth plate phenotype. Actually, because of abnormal deaths of the hypertrophic chondrocytes, there is a marked loss of the hypertrophic zone which becomes fully mineralized. This can be shown by demineralization of the vertebral sections (**Fig. 5B**) and also by Type X collagen immunofluorescence (**Fig. 5E**).

8. The collagen X antibody is dreadful, which is there intracellular staining? Nothing can be interpreted from this experiment.

Answer: We agree; we now provide new immunofluorescence images generated by using a new anti-type X collagen antibody (**Fig. 5E**).

9. I have quite a few problems with Figure 6. I am surprised that the authors could not purchase a good MGP antibody, there are lots on the market and not being able to use one to study the mouse model is a weakness of the study. To overcome this, they expressed WT and mutant MGP in cells.

Answer: We regret to inform the reviewer that so far, we have examined multiple commercial antibodies, but unfortunately all of them detected MGP protein in extracts from MGP null mice in which the whole MGP gene has been deleted (do not produce MGP mRNA). Please see the **Fig. 2 for Reviewers** (see at the end of this document). Due to the lack of a suitable antibody that can detect the mouse MGP protein we relied on tagged MGP proteins that were expressed in various cell culture experiments.

10. In figure 6D and E, what is the 35 kDa band? – I can't find any reference that MGP dimerises – so can it be an artefact?

Answer: We predict that the high molecular weight band represents dimerized MGP as this band disappeared when we reduced the protein amount and increased the amount of 2-Mercaptoethanol while denaturing samples for Western blotting. The original Western blotting images are now replaced (**Fig. 7D and E**).

11. Furthermore, in Fig6D a large proportion of the mutant MGP appears to be processed correctly - i.e. correctly cleaved signal peptide to give the 15 kDa band – how do the authors reconcile this?

Answer: Our explanation for this is that the C19F mutation does not prevent the cleavage of the signal peptide completely, rather a significant fraction of the mutated proteins are not cleaved. This explanation was supported by our mass spec analyses.

12. In addition, it is the unprocessed (17 kDa band) species that is secreted in the mutant cell model – please explain? Why is this 17 kDa band not detected by mass spec?

Answer: Although in lesser amount, a part of the unprocessed protein is secreted as well. Please note that for the mass spec analyses, both the bands corresponding the unprocessed and processed proteins were cut out and digested by trypsin. Our analyses confirmed that C19F MGP is not always cleaved at the cystine 19 residue. It is not unusual that the full-length signal peptide is not detected by mass spec analyses.

13. The authors should also run these gels unreduced because the mutant MGP should have an additional unpaired cysteine, which may cause the mutant MGP to aggregate.

Answer: It is possible that the unpaired cysteine may cause aggregation. However, considering that in some of our previous Western blotting experiments insufficient denaturation of proteins showed multimerization of native MGP without the presence of the C19F mutation, we believe that Western blotting under denaturing condition will not provide a conclusive answer.

14. The authors hypothesis that mutant MGP is retained within the ER and causes stress – this is where a good MGP antibody and in vivo validation would have been ideal. The authors only look at CHOP and calnexin as ER-stress markers (Fig. 7). They need to examine markers for all branches of the UPR – I would suggest BiP and Xbp1 splicing, ATF6 cleavage and phosphorylation of eif2a.

Answer: As mentioned above, we have no good anti-MGP antibody at our disposal. However, with anti-FLAG antibody we showed FLAG-tagged mutant MGP protein was co-localized in ER with ER marker calnexin (**Fig. 8C**). **Also**, as suggested by the reviewer, we have now included a thorough examination of the markers ATF6-N, phosphorylated eIF2 α (peIF2 α) and XBP1 for all three ER stress pathways *in vivo* (**Fig.9**).

15. The authors next attempted to validate ER-stress in the mouse model (Figure 8). Unfortunately, the calnexin antibody is not good enough – why all the extracellular staining? It’s just not specific and no conclusions can be drawn from this experiment. Furthermore, why is DAPI staining the bone matrix, particularly in the mutant – that should not be happening. In the hypertrophic cells (shown by red arrows in mutant) the DAPI appears to be staining the lacunae occupied by the cell and not nucleus at all. This need to be repeated before any meaningful interpretation can be made.

Answer: We have repeated the anti-calnexin immunostaining experiments and re-taken the images by using the confocal microscope. The calnexin staining is primarily intracellular as expected. DAPI staining is nuclear in all the new images (**Fig. 9A**).

16. I’m not convinced by the EM image (there is also no wt control) those large areas don’t look like distended ER to me – there are no ribosomes.

Answer: Since we have now performed more thorough *in vitro* (using 4 PBA) and *in vivo* analyses of ER stress markers, we omitted the EM image included in the initial version of the manuscript.

17. In figure 8C I’m not convinced the images are correctly labelled – to me they show rounded resting zone chondrocytes – there seems to be little evidence of stacked proliferative zone chondrocytes.

Answer: We thank the reviewer for raising this concern. This has been corrected in the revised version. Also, please note that the regular cellular and tissue structure in the mutant growth plate is altered and may not appear the same as a wild type growth plate.

Reviewer #3:

Matrix Gla protein encoded by MGP is a secreted inhibitor of calcification in the extracellular matrix. MGP also negatively regulates BMP activity. Over expression of MGP decreases mineralization, delays chondrocyte maturation and inhibits endochondral ossification (PMID: 10579728) while loss of function causes abnormal excessive vascular calcification and premature bone mineralization, reduced bone growth and osteopenia (review PMID: 16612082). It is therefore an important regulator of physiological mineralization. Mutations in MGP cause Keutel syndrome (KS) an autosomal recessive disease affecting the skeleton such as hands, and limbs and midface malformation and in which there is abnormal increase in cartilage and arterial calcification. Other abnormalities occur, such as hearing loss, congenital heart defect. In this paper Gourgas et al describe 4 affected spondyloepiphyseal skeletal dysplasia individuals with heterozygous mutations in MGP that alter Cys19 to phe or tyr.

This newly described spondyloepiphyseal dysplasia is interesting because it is an autosomal dominant mutation in MGP, of unknown etiology. In the paper the authors generate a mouse model that

recapitulates the human phenotype. Because the mutation affects the signal peptide the authors hypothesize that the mutation will affect the secretion of MGP and trigger ER stress. They test this hypothesis in cells and the mouse model. The study is important because it could potentially shed new insights into the role of MGP, the potential dominant negative molecular mechanism underlying heterozygosity for the MGP Cys19 mutation. It also could extend the examples in which skeletal dysplasias are caused by the impact of ER stress on chondrocyte differentiation and bone formation. However, while the study has potential interest, the major issue is that the depth of investigation is insufficient to define clearly the underlying molecular mechanism(s) and explain how heterozygosity for the MGP. Cys19 affects chondrocyte differentiation and cause the bone defects as outlined below. The role of ER stress in the underlying molecular pathogenesis is incompletely established.

We thank the reviewer for finding our manuscript potentially interesting. A point-by-point response has been included below.

Major concerns:

1. One major concern is that the characterization of the phenotype is not in sufficient depth. The study of the changes in humans and mice begin after the phenotype has manifested – the youngest patient is 2 years 4 months and at that age the epiphyses is already smaller; the earliest stage the Mgp 56G>T mice are analysed for skeletal defects e.g. bone lengths and bone mass is at 6 weeks – that is equivalent to a young human adult.

Answer: We start observing the difference in body weight between the control and mutant mice starting from 2 weeks onward. Following the reviewers' suggestions, we have performed additional bone analyses. We now include micro-CT data showing that long bone length and bone mass are virtually indistinguishable at 1 and 2 weeks of age between these two genotypes (**Fig.2F**). We have now performed more thorough analyses of ER stress markers in the 3-week-old growth plates of control and mutant mice (**Fig.9**). We observe massive apoptosis of growth plate chondrocytes at 3 weeks of age (**Fig.10A**). Taken together, our data suggest that the skeletal traits of the mutant mice start appearing after 2 weeks of age. Since we cannot breed these mice, and direct microinjection or IVF are the only ways to propagate these mice, we are unable to generate a large number of mice for analyses at multiple time points. This is why we decided to examine the bones thoroughly at 6 weeks of age only.

2. The age of mice analysed in Fig. 4 and Fig. 8 are not stated.

Answer: We have corrected this oversight in the revised version.

3. In Fig. 5, 3 week (equivalent to adolescent human) and 6 weeks mice are described. The stage and onset of the abnormal phenotype should be characterized especially since Mgp is already expressed in the developing cartilage at E14.5.

Answer: The reviewer is right, MGP is expressed early as shown by gene expression analyses initially performed by Dr. Karsenty's group. However, as mentioned in the answer to comment 1, our data

indicates that the skeletal abnormalities appear post 2 weeks of age. Nevertheless, we included additional data of early age bone analyses (**Fig.2F**).

4. In addition, a prerequisite of the study must be characterization of the expression of *Mgp* 56G>T/MGPCys19. In which cells in the developing bone is the mutant expressed. Is it expressed in perichondrium/periosteum? In osteocytes? This has not been carried out.

Answer: Currently, there is no antibody available that can detect the mutant MGP protein specifically. It is highly unlikely that the point mutation in MGP's signal peptide will affect its mRNA expression pattern and we expect a similar expression pattern to that of the wild type gene. As published by Luo et al (<https://asbmr.onlinelibrary.wiley.com/doi/abs/10.1002/jbmr.5650100221>) in the growth plates of the developing bones, *Mgp* mRNA is present in resting, proliferative, and late hypertrophic chondrocytes. The same paper reports that there is no expression of *Mgp* in the perichondrium, periosteum or osteoblasts.

5. The authors have not distinguished sufficiently between whether MGP Cys19 causes complete loss of function of MGP vs an impact of MGP Cys19 on wild-type MGP vs a neomorphic impact. That is, if or how the mutant affects the normal allele. Is the mutation effectively creating a null or has a dominant negative/new effect?

Answer: This is a very interesting point. The mutation is not a null mutation, at least not in all the cells expressing the mutant protein, as there is no vascular calcification in the heterozygote mice expressing C19F MGP (Note, global MGP-deficient mice develop widespread vascular calcification). Also, our new experimental data shows the growth plate features and cell phenotypes are very different between MGP-null and the mutant heterozygote *Mgp*^{+/^{56G>T}} mice (See **Fig. 5A and 10C**). Although our new data suggests that the accumulation of the mutant protein in the endoplasmic reticulum (ER) may also affect the secretion of the wild type protein (**Supplemental Fig. 6**), but this is most likely a general effect of ER stress. We believe that the growth plate phenotypes start with the accumulation of the mutant protein leading to ER stress and apoptosis. This results in a marked decrease of secreted MGP levels caused by both the loss of MGP producing cells as well as ER stress-associated impaired release of the wild type protein. Collectively, our findings suggest that it is a new gain of function trait caused by abnormal accumulation of the mutated protein that in turn initiates the growth plate phenotype; however there might be some additive effects of reduced presence of wild type MGP in the growth plate extracellular matrix. We have now highlighted this in the discussion.

6. In mice loss of MGP function results in increased calcification. In the MGP Cys19 patient 4 premature calcification is reported. Low bone mass and decreased mineralization is reported for *Mgp* 56G>T heterozygote mice. A comparison of the *Mgp* 56G>T heterozygote with the homozygous null would be informative.

Answer: The generation of a homozygous mice for *Mgp* 56G>T (MGPC19F) mutation is an excellent suggestion. However, since the heterozygote mice are unable to breed, it would be difficult if not impossible to generate homozygous null mice within the scope of the current study.

7. Further it would be important to compare the phenotypes of homozygous *Mgp* 56G>T and also compound *Mgp* 56G>T /null, to assess dominant interference with wild-type MGP.

Answer: Thank you for this suggestion, but as stated above, we are unable to generate homozygous mice at this point.

8. Why is there less bone? The phenotype characterization of the chondrocyte differentiation defect and possible impact on osteoblast differentiation resulting in less bone, are superficial, with inadequate use of key molecular markers to pinpoint what phase of chondrocyte or osteoblast differentiation is affected. Using alcian blue staining (for proteoglycans in cartilage) is hardly sufficiently specific to determine whether late hypertrophic chondrocyte differentiation is affected. Indeed in Fig. 6 A and B only histological staining is used. In those sections the von kossa staining obscures assessment of the chondrocyte morphology. Little information can be surmised on any impact on osteoblast differentiation.

Answer: Although relatively new, transdifferentiation of hypertrophic chondrocytes to osteoblasts is now considered as a major pathway for trabecular bone formation. Several key studies using cutting age genetic tagging and imaging methodologies have confirmed that while some hypertrophic chondrocytes undergo apoptosis, a large number of these cells undergo transdifferentiation to become osteoblasts to form the trabecular bones. In our mutant model these hypertrophic cells are undergoing apoptosis thereby depleting a critical source for continuous supply for osteoblasts, and thereby reducing their number and causing a low bone mass phenotype. Following the reviewer's suggestion, we have now provided new data suggesting that the progression of chondrocyte differentiation in the growth plate is not inhibited as evident by type II collagen and aggrecan expression (**Fig.5C and D**), although the expression of both markers appear to be increased. Since *Mgp* is not expressed by osteoblasts in vivo, we did not examine the effect of the novel mutation on osteoblast differentiation.

(<https://asbmr.onlinelibrary.wiley.com/doi/abs/10.1002/jbmr.5650100221>) in

9. Tibia and vertebra are shown in Fig 5 A and B, but the end-plate and growth plate of the IVD are not clearly shown.

Answer: We have now included the better images of long bone and vertebral growth plates in the **Supplemental Fig.5** and **Fig. 1 for Reviewer** (see at the end of this document).

10. There also seems to be a difference in the morphology of the nucleus pulposus but that is not mentioned or investigated.

Answer: The reviewer is correct; indeed the nucleus pulposus is morphologically somewhat different in *Mgp*^{+/56G>T} mice with higher number of cells and more extracellular matrix. Currently, we do not have an explanation for this trait. We have added this in the discussion section in our revised manuscript.

11. Fig 5C shows type X collagen staining but it is of the 3-week vertebrae, and are these sections of the growth plate or endplate?

Answer: The staining shows the growth plate. This information is now added to the Figure legend.

12. And what is the impact on both those structures as chondrocyte hypertrophy is involved in both. And the impact on the secondary ossification center should be described too.

Answer: We have now discussed both the growth plate and end plate structures in the *Mgp*^{+/^{56G>T}} mice expressing C19F MGP in the discussion. Since we do not see any difference in the mutant bones for up to 2 weeks of age, we concluded that the early-stage development of the secondary ossification centers that is formed after birth are not affected.

13. The authors then go on to test the hypothesis that inability to secrete MGP C19F triggers ER stress causing apoptosis of chondrocytes and possibly their differentiation to osteoblasts. However, since there are 2 sources of osteoblasts – the periosteum and the hypertrophic chondrocytes it could well be an impact on periosteal osteoprogenitors as well.

Answer: The reviewer is correct about the two sources of osteoblasts. While both periosteum and hypertrophic chondrocytes serve as sources for osteoblasts during the embryonic and perinatal stages, overtime periosteal contribution is diminished and hypertrophic chondrocytes serve as the main source of osteoblasts. Therefore, in endochondral bones the remodeling activities of trabeculae are primarily localized close to the growth plates, not so much in the diaphysis region. Also, an important point should be noted here that osteoblasts do not express *Mgp* (<https://pubmed.ncbi.nlm.nih.gov/7754814/>), rather MGP is produced by the prehypertrophic and hypertrophic chondrocytes in the growth plates. Therefore, it is really less likely that there will be any accumulation of mutant MGP in the periosteal osteoblasts or osteoblasts at any other location. Indeed, the absence of any visible skeletal abnormality before two weeks of age in the mutant mice supports the above inference.

14. Given that many papers and reviews are available on the impact of ER stress on chondrocyte differentiation in vivo and in vitro (e.g. in ATDC5 cells) (e.g. PMID: 33307190, PMID: 34297411, PMID: 30902258, PMID: 27002737) the authors should have included more convincing data and insights into which arm of the UPR/ER stress signaling is activated that could explain the chondrocyte differentiation defect.

Answer: Thank you for this very important point. We have included more thorough analyses of different arms of ER response pathways (**Fig. 9**).

15. Furthermore, given that several papers have been published showing the feasibility of overcoming at least partially the impact of the ER stress using pharmacological methods e.g. carbamazepine (CBZ), ISRIB and 4PBA (PMID:32399188, PMID: 28920921, PMID: 30024379, PMID: 34990412), it would be important to demonstrate some rescue effects of either inhibiting the UPR or promoting proteostasis to reduce intracellular accumulation, at least in cell culture.

Answer: Once again thank you for this very insightful comment. We have now provided new experimental data using chondrogenic ATDC5 cells which matches to our previous findings from HEK293 cells. More importantly, we now show that 4PBA, an inhibitor of ER stress/unfolded protein response can prevent abnormal cell deaths caused by C19F MGP expression in ATDC5 cells (**Fig. 10D, E and F**). This new finding will be the basis for future experiments to prevent the initiation and progression of the skeletal pathologies in mouse models carrying the G56>T mutation which in turn may have a significant impact on the future treatment strategy for humans carrying this mutation.

16. Overexpressing MGP in the developing limb has been shown to delay chondrocyte maturation and inhibit cartilage mineralization (PMID: 10579728). To what extent is the impact of MGP Cys19 due to increased stability/persistence of the protein versus triggering ER stress?

Answer: This is an interesting question; however we do not think that the increased stability of the mutant protein is causing the growth plate phenotype. Increased MGP would have led to poor mineralization of the growth plate cartilage. In the contrary, the mineralization of the hypertrophic zone is not reduced as seen in the *in vitro* limb culture experiments described above. If anything, there is a more continuous mineralization of the growth plate, particularly in the prehypertrophic/hypertrophic zones.

17. The authors have previously shown in *Mgp* null mice that there was increased apoptosis in chondrocytes. To what extent is the apoptosis triggered in MGP Cys19 due to loss of function versus or the activation of the UPR?

Answer: Thank you for following our work on MGP biology, we really appreciate this recognition of our previous work! Our previous apoptosis analyses were performed on the cartilage of the nasal septum. Here for the first time, we performed new experiments comparing apoptosis in MGP null (*Mgp*^{-/-}) and the new *Mgp*^{+/^{56G>T}} mice expressing C19F MGP. The growth plates of our new mutant mice showed markedly increased TUNEL positive cells in comparison to that of MGP null growth plates (**Fig. 10C**).

18. In that regard the characterization of UPR markers *in vivo* in the mutants was rather superficial. Chop has multiple functions that can be anti-apoptotic or pro-apoptotic and can be activated by other factors not exclusively by the UPR. It would also have been more convincing if a chondrocyte progenitor cell line (e.g. ATDC5) has been used to demonstrate the accumulation of MGP Cys19 in the ER and impact on differentiation *in vitro*, rather than kidney cells.

Answer: Thank you for this suggestion; in addition to more thoroughly performed *in vivo* ER stress analyses (**Fig. 9**), we have now included data from ATDC5 cells (**Fig. 6, 10D, E and F**) which matches the findings from our experiments on HEK293 cells.

19. Furthermore, co-staining of the mutant protein with an ER marker would provide stronger evidence for accumulation in the ER.

Answer: We have included new images showing the co-localization of the mutant protein (C19F MGP) together with calnexin confirming that the mutant protein is actually present in the ER (**Fig. 8C**).

20. Would the authors discuss what is the implication of a lack of aorta calcification in the *Mgp* 56G>T mice?

Answer: The lack of arterial calcification in *Mgp*^{+/^{56G>T} mice implies that unlike *Mgp*^{-/-} mice, these mice do not die due to arterial rupture. Since one allele is still functional and a fraction of the mutated protein can also be processed for signal peptide cleavage, we believe that there will be sufficient functional MGP in the vascular matrix to prevent ectopic calcification. This later conclusion implies that the mutation may differentially affect various cell types expressing MGP.}

Figures for Reviewers (not for publication)

Figure 1 for Reviewers: Abnormal growth plates in *Mgp*^{+/*56G*>*T*} mice. A and B. Von Kossa and van Gieson (VKVG) staining of lumbar vertebra sections from 6-week-old *Mgp*^{+/*56G*>*T*} mice. The growth plates of different *Mgp*^{+/*56G*>*T*} mice are prematurely closed with abnormal mineral deposition. We included sections from 3 different mutant mice to demonstrate the presence of same growth plate features and to rule out any possible sectioning artifact.

Figure 2 for Reviewers: Unsuccessful attempts to use two anti-mouse MGP antibodies in Western blotting. Our repeated efforts to detect mouse MGP protein by Western blotting were unsuccessful. Tissues extracts from various mouse models using anti-MGP antibodies from a collaborator (A), or a commercial vendor (B) were analyzed. Note that in the trachea extracts, a 14 kDa band (approximately) appears in all the samples including the one from *Mgp*^{-/-} mouse in which *Mgp* gene is completely ablated (does not produce any MGP protein). When aorta extracts were examined by Western blotting using an anti-MGP antibody from a vendor, MGP protein was not detected in wild type or any other genotypes. The red boxes identify the samples from mutant mice lacking MGP, while the blue boxes show the area of the blots where bands corresponding to MGP are expected.

Figure 3 for Reviewers: Effects of conditional ablation of *Ggcx* gene in chondrocytes. *Ggcx^{flox/flox}* mice were mated with *Col2a1-Cre* mice to generate *Ggcx^{flox/flox};Col2a1-Cre* mice. Although the long bones in these mice are shorter, they show an overall milder skeletal phenotypes than *Mgp^{+/56G>T}* mice. The percent reduction of tibial and femoral lengths are larger in *Mgp^{+/56G>T}* mice than that of *Ggcx^{flox/flox};Col2a1-Cre* mice. The growth plate pathology in *Ggcx^{flox/flox};Col2a1-Cre* mice is milder (more like in *Mgp^{-/-}* mice) in comparison to that of *Mgp^{+/56G>T}* mice (unpublished data).

REVIEWERS' COMMENTS

Reviewer #1 (Remarks to the Author):

The response to the reviewers comments were adequately addressed. Yet, some questions still remain as the precise mechanism by which the C19F mutation in MGP causes premature death (in mice) is still not known. The authors respond that no vascular calcification is seen, and thus involvement of the vascular system in premature death is unlikely. In here, the authors assume that the function of MGP is purely as anti-mineralisation protein. The fact that for the human MPG knock-out only 40 odd cases are known points towards a developmental role for MGP. Indeed, MGP can bind BMP2/4 as well as is involved in arteriovenous malformations. Also here, absence or immaturity of blood vessels in the bone might causative for decreased mineralisation.

The authors need to take these points into consideration, also - as the authors state correctly themselves - men are no mice and visa versa.

Reviewer #2 (Remarks to the Author):

The authors have considerably revised this manuscript and have provided additional experimental data to clarify and address my concerns in the original submission. These substantial revisions have made a stronger paper.

Reviewer #3 (Remarks to the Author):

In this revised manuscript the authors have addressed many of the questions raised and significantly improved the phenotype analyses of the of MGP Cys19 mutants, demonstrated that ER stress is triggered and the data quality is much better. The paper's major novelty is identifying a new subtype of chondrodysplasia that is caused by the Cys19 mutation in MGP

However there are still some questions outstanding about the underlying mechanism behind the chondrodysplasia. The comparison with different mutant alleles of MGP attempt to examine the possibility of dominant negative impact and the presence of phenotype in heterozygotes are consistent with dominant negative mechanism. However molecular explanation /data supporting the dominant negative mechanism is still incomplete. The question about "to what extent is the impact of MGP Cys19 due to increased stability/persistence of the protein versus triggering ER stress?" was aimed at obtaining

more insights into the possibility of a dominant negative effect of the Cys19 mutation. Furthermore an explanation of the growth plate and bone defects and the impact on chondrocyte differentiation is still lacking. While it is interesting that apoptosis was increased it may not explain the skeletal defect. For example is hypertrophic differentiation affected? It is noticeable that there were significantly fewer cells expressing collagen X but perhaps more expressing COL2A1 in the hypertrophic zone (Fig . 5C,E). It has been shown that ER stress affects the differentiation progression of hypertrophic chondrocytes (PMID: 30024379 PMID: 34990412 PMID: 32633442). Markers probing the differentiation progression and status of the chondrocyte would be informative.

Now the paper is emphasizing the impact of ER stress triggered by the MGP Cys19, one would have expected much better discussion of the phenotypic and molecular similarities and differences compared to other skeletal dysplasias associated with ER stress and studies with chemical chaperones, inhibitors of the PERK pathway or promoting protein degradation (PMID: 30024379 PMID: 34990412 PMID: 32633442 PMID: 28920921). There is a wealth of literature on the impact of ER stress on chondrocyte differentiation and approaches at rescuing the defect including clinical trials (CBZ) in humans. Yet there is little in the way of discussion on these important points in the revised manuscript.

Other points:

Detection of the spliced form of Xbp1 mRNA is diagnostic of the UPR not detection of XBP1 alone.

RESPONSE TO REVIEWERS' COMMENTS

We thank the reviewers for carefully reviewing our revised manuscripts. We have further modified the text file of our manuscript to address the new comments from the reviewers. A point-by-point response is provided below:

Reviewer #1 (Remarks to the Author):

The response to the reviewers comments were adequately addressed. Yet, some questions still remain as the precise mechanism by which the C19F mutation in MGP causes premature death (in mice) is still not known. The authors respond that no vascular calcification is seen, and thus involvement of the vascular system in premature death is unlikely. In here, the authors assume that the function of MGP is purely as anti-mineralisation protein. The fact that for the human MPG knock-out only 40 odd cases are known points towards a developmental role for MGP. Indeed, MGP can bind BMP2/4 as well as is involved in arteriovenous malformations. Also here, absence or immaturity of blood vessels in the bone might causative for decreased mineralisation. The authors need to take these points into consideration, also - as the authors state correctly themselves - men are no mice and visa versa.

Our response: The reviewer is correct that human pathologies caused by MGP deficiency shows a role for this protein in the skeletal development. While calcification of the cartilaginous tissues, particularly of the growth plates, is considered to be the main cause, other independent cellular alterations e.g., differentiation abnormalities and apoptosis cannot be ruled out. In case of our heterozygote mouse model expressing the C19F variant of MGP, we see abnormal ectopic calcification of the growth plate, but also a very high rate of chondrocyte apoptosis which is caused by endoplasmic reticulum (ER) stress due to the accumulation of the mutant protein in the ER. In addition, some increase of chondrocyte markers, e.g., type II collagen and aggrecan were also noticed in the growth plates of the mutant mice. Collectively, the phenotype appears to be a complex one mainly caused by chondrocyte apoptosis together with impaired inhibition of calcification of the growth plate matrix and possibly by other yet unknown cellular alterations. We have now added these possibilities in the discussion of the revised manuscript.

Regarding 'decreased mineralisation,' we would like to point out that we have evidence for a poor bone mass in the mutant mice, but we did not notice osteomalacia as evident by histological staining. In fact, mineralization is increased in the growth plate.

Reviewer #2 (Remarks to the Author):

The authors have considerably revised this manuscript and have provided additional experimental data to clarify and address my concerns in the original submission. These substantial revisions have made a stronger paper.

Our response: We thank the reviewer for the valuable comments made to improve the quality of our manuscript.

Reviewer #3 (Remarks to the Author):

In this revised manuscript the authors have addressed many of the questions raised and significantly improved the phenotype analyses of the of MGP Cys19 mutants, demonstrated that ER stress is triggered, and the data quality is much better. The paper's major novelty is identifying a new subtype of chondrodysplasia that is caused by the Cys19 mutation in MGP. However, there are still some questions outstanding about the underlying mechanism behind the chondrodysplasia. The comparison with different mutant alleles of MGP attempt to examine the possibility of dominant negative impact and the presence of phenotype in heterozygotes are consistent with dominant negative mechanism. However molecular

explanation /data supporting the dominant negative mechanism is still incomplete. The question about “to what extent is the impact of MGP Cys19 due to increased stability/persistence of the protein versus triggering ER stress? “was aimed at obtaining more insights into the possibility of a dominant negative effect of the Cys19 mutation. Furthermore an explanation of the growth plate and bone defects and the impact on chondrocyte differentiation is still lacking. While it is interesting that apoptosis was increased it may not explain the skeletal defect. For example is hypertrophic differentiation affected? It is noticeable that there were significantly fewer cells expressing collagen X but perhaps more expressing COL2A1 In the hypertrophic zone (Fig . 5C,E). It has been shown that ER stress affects the differentiation progression of hypertrophic chondrocytes (PMID: 30024379 PMID: 34990412 PMID: 32633442). Markers probing the differentiation progression and status of the chondrocyte would be informative. Now the paper is emphasizing the impact of ER stress triggered by the MGP Cys19, one would have expected much better discussion of the phenotypic and molecular similarities and differences compared to other skeletal dysplasias associated with ER stress and studies with chemical chaperones, inhibitors of the PERK pathway or promoting protein degradation (PMID: 30024379 PMID: 34990412 PMID: 32633442 PMID: 28920921). There is a wealth of literature on the impact of ER stress on chondrocyte differentiation and approaches at rescuing the defect including clinical trials (CBZ) in humans. Yet there is little in the way of discussion on these important points in the revised manuscript.

Other points:

Detection of the spliced form of Xbp1 mRNA is diagnostic of the UPR not detection of XBP1 alone.

Our response: We thank the reviewer for his efforts and suggestions to further strengthen the quality of our manuscript. We believe that while there is a dominant negative angle (e.g. premature ectopic calcification of the growth plate) explaining the observed skeletal traits, the key reason underlying the chondrodysplasia is abnormally high apoptosis of the growth plate chondrocytes which is caused by ER stress due to impaired secretion and accumulation of C19F MGP in the ER. The loss of the hypertrophic chondrocytes en masse results in the severe growth plate anomalies impairing the growth of the bones and other skeletal pathologies. Further, the loss of the hypertrophic chondrocytes, an important source of osteoblast precursors in mice is the primary cause of the low bone mass in the mutant mice. We have discussed these possibilities in our revised manuscript to explain the skeletal pathologies seen in mice and humans affected by the expression of the C19F variant of MGP.

We thank the reviewer for asking us to include further discussion on other skeletal dysplasias associated with ER stress and studies with chemical chaperones. Accordingly, we have modified the discussion of the revised manuscript to integrate these suggestions.

We apologize for our oversight as initially we did not indicate which form of XBP1 was detected by our antibody. Indeed, we used an antibody that detects the spliced form of XBP1 (sXBP1) in our study. This mistake has been corrected in the revised manuscript.